# FlashDP: Memory-Efficient and High-Throughput DP-SGD Training for Large Language Models

## Abstract

As large language models (LLMs) increasingly underpin technological advancements, the privacy of their training data emerges as a critical concern. Differential Privacy (DP) serves as a rigorous mechanism to protect this data, yet its integration via Differentially Private Stochastic Gradient Descent (DP-SGD) introduces substantial challenges, primarily due to the complexities of per-sample gradient clipping. Current explicit methods, such as Opacus, necessitate extensive storage for per-sample gradients, significantly inflating memory requirements. Conversely, implicit methods like GhostClip reduce storage needs by recalculating gradients multiple times, which leads to inefficiencies due to redundant computations. This paper introduces FlashDP, an innovative cache-friendly method that consolidates necessary operations into a single task, calculating gradients only once in a fused manner. This approach not only diminishes memory movement by up to **50%** but also cuts down redundant computations by **20%**, compared to previous methods. Consequently, FlashDP does not increase memory demands and achieves a **90%** throughput compared to the Non-DP method on a four-A100 system during the pre-training of the Llama-13B model, while maintaining parity with standard DP-SGD in terms of accuracy. These advancements establish FlashDP as a pivotal development for efficient and privacy-preserving training of LLMs.

## 1 Introduction

The transformer architecture (Vaswani et al., 2017) has revolutionized fields like natural language processing (Gao et al., 2024; Xie et al., 2023), embodied AI (Song et al., 2023; Duan et al., 2022; Xu et al., 2024), and AI-generated content (AIGC) (Cao et al., 2023; Wu et al., 2023), with Large Language Models (LLMs) demonstrating exceptional abilities in text generation, complex query responses, and various language tasks due to training on massive datasets. These models, exemplified by ChatGPT, are applied across diverse areas, including healthcare, where they enhance diagnosis and drug discovery by analyzing medical data (Toma et al., 2023; Ali et al., 2023; Sheikhalishahi et al., 2019; Sallam, 2023; Biswas, 2023). However, the extensive capabilities of LLMs raise significant privacy concerns, particularly as they can inadvertently expose or generate sensitive information, owing to their potential to memorize data from large training sets (Pang et al., 2024; Nasr et al., 2023; Carlini et al., 2023; Ippolito et al., 2022; McCoy et al., 2023; Tirumala et al., 2022; Zhang et al., 2023; Ashkboos et al., 2023).

Differential Privacy (DP) ensures privacy by adding noise during data processing, such that any single data point's influence on outcomes is minimal (Dwork, 2006). As the most commonly adopted methods for ensuring DP in deep learning models, Differentially Private Stochastic Gradient Descent (DP-SGD) based methods (Abadi et al., 2016) adapt traditional stochastic gradient descent by clipping gradients per sample and adding noise. Although DP-SGD's application in LLMs is increasing, recent research (Li et al., 2022; Bu et al., 2023; Anil et al., 2022; Hoory et al., 2021) primarily targets the fine-tuning phase, providing privacy only for fine-tuned data. While some studies (Lee & Kifer, 2021; Li et al., 2022; Bu et al., 2023) have applied DP-SGD to pre-training, they typically use shorter sequence lengths, not maximizing the benefits of longer sequences used in modern LLMs. This limitation arises from the high computational and memory demands of DP-SGD, especially with long sequences typical in LLM pre-training.

Integrating DP into LLM training via DP-SGD/Adam poses significant challenges, particularly due to per-sample gradient clipping. This crucial privacy technique involves adjusting each data sample's gradients to limit their influence on model updates. While critical for maintaining strict privacy standards, this approach requires computing and storing individual gradients, significantly raising computational and memory demands. Managing these gradients is especially taxing in LLMs, which are known for their large parameter spaces. Each gradient must be carefully clipped and aggregated before updating model parameters, straining computational resources, and prolonging training times. These scalability issues are particularly acute in settings with limited hardware, creating significant barriers to efficiently training privacy-aware LLMs (Li et al., 2022; Bu et al., 2023).

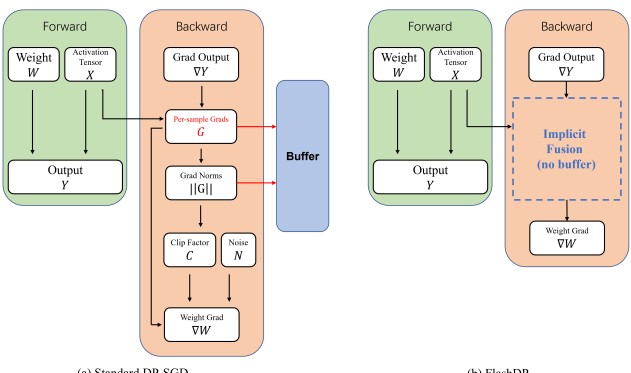

Figure 1: Comparison of different training methods. (a) Standard DP-SGD: Stores per-sample gradients **G** (red explicit cache), increasing memory usage (blue buffer). (b) FlashDP: Optimizes gradient processing by consolidating computations into a single pass, reducing redundancy and memory use.

Current research on DP-SGD for training LLMs can be categorized into two classes: explicit methods like Opacus (Yousefpour et al., 2021) stand out by directly storing per-sample gradients. This approach, while straightforward, significantly increases the memory footprint (Appendix Table 4), which becomes prohibitive for state-of-the-art LLMs characterized by billions of parameters (Touvron et al., 2023; Achiam et al., 2023). Such a substantial increase in memory requirements hampers scalability and renders these methods impractical for deployment in large-scale model training environments. The direct storage of gradients, essential for ensuring the privacy guarantees of DP, thus poses a substantial barrier to the efficient implementation of DP in LLMs.

Conversely, implicit methods, exemplified by innovations such as GhostClip (Li et al., 2021), address the memory challenge by circumventing the need for persistent storage of per-sample gradients. These methods segment the DP-SGD process into multiple discrete computational tasks, ostensibly to mitigate memory demands. However, this strategy necessitates the frequent recalculation of per-sample gradients, which introduces a high degree of computational redundancy (Table 4). This redundancy not only undermines training efficiency but also extends the duration of the training process significantly. For LLMs, which require substantial computational resources and extended training times, the inefficiencies introduced by such redundant computations become a critical bottleneck. These implicit methods, while innovative in reducing memory usage, thus struggle to deliver a practical solution for the privacy-preserving training of LLMs at scale.

To effectively tackle the challenges presented by existing methods of integrating DP into the training of LLMs, we introduce FlashDP, a novel, cache-friendly implicit algorithm designed to streamline the DP-SGD process (Figure 1 (a)). FlashDP uniquely implements a unified computational strategy that performs the gradient operations required for DP-SGD in a single pass (Figure 1 (b)). This innovative approach not only eliminates the need for multiple recalculations of per-sample gradients but also consolidates the entire process into one cohesive computational task. To be specific, FlashDP's architecture, which consolidates the entire DP-SGD process into a single GPU kernel, eliminates redundant computations and optimizes data flow within the GPU. This integration results in a streamlined workflow that efficiently manages memory and processing resources. Also, FlashDP reorganizes the GPU operations to maximize data throughput and minimize latency, effectively enhancing the overall efficiency of the training process. These architectural improvements significantly reduce the volume of memory transfers and computational redundancies, thereby optimizing both the speed and resource utilization during the training of LLMs with DP.

By re-designing the gradient computation workflow, FlashDP dramatically reduces the volume of memory transfers by 50% and decreases redundant computational tasks by 20% compared to previous implicit methods. This optimization is achieved through an advanced caching mechanism

that efficiently manages gradient data and computation within GPU memory, minimizing the data movement across the system. As a result, FlashDP significantly alleviates the memory overhead traditionally associated with DP-SGD, enhancing the model's scalability and training speed.

The practical impact of these improvements is substantial. On a computational platform equipped with four NVIDIA A100 GPUs, FlashDP achieves a remarkable 90% throughput compared to the non-DP method during the pre-training phase of the Llama-13B model, a state-of-the-art LLM known for its extensive data and computation demands. Crucially, this enhanced performance is attained without any degradation in the accuracy or dilution of the privacy guarantees that are fundamental to DP-SGD. FlashDP thus not only meets but exceeds the operational requirements for effective and efficient privacy-preserving training of LLMs.

Our contributions can be summarized as follows:

- **Enhanced Throughput for Long Sequence LLM training with DP**: We propose FlashDP, which effectively resolves the issue of low throughput in DP-SGD/Adam during the training of LLMs with long sequence lengths. By optimizing the computational workflow and integrating more efficient handling of per-sample gradients, FlashDP significantly enhances the processing speed without compromising the model's accuracy or privacy integrity.

- **Innovative GPU I/O Optimization**: Our study pioneers the exploration of DP-SGD from the perspective of GPU input/output operations. FlashDP's architecture, which consolidates the entire DP-SGD process into a single GPU kernel, eliminates redundant computations and optimizes data flow within the GPU. This approach not only reduces the computational load but also minimizes the number of GPU memory accesses, setting a new standard for efficiency in DP implementations.

- **Experimental Validation of Efficiency and Scalability**: In practical LLM models involving Llama-13B, FlashDP matches the speed and memory usage of non-DP training methods and achieves a significant **90%** throughput compared with Non-DP methods. This performance is achieved on a computational platform equipped with four NVIDIA A100 GPUs. Importantly, it accomplishes this without any degradation in the precision or the privacy guarantees typically observed in standard DP-SGD implementations. This capability demonstrates FlashDP's effectiveness in scaling DP applications to larger and more complex LLMs without the usual trade-offs.

## 2 RELATED WORK

**Improving Time and Memory Complexities of DP-SGD.** The transition from standard stochastic gradient descent to DP-SGD introduces substantial modifications in memory and computational demands. In conventional settings, parameter updates are efficiently computed by aggregating gradients across all samples within a batch. This approach is both memory-efficient and computationally straightforward. In contrast, DP-SGD mandates that each sample's gradients be preserved, clipped, and subsequently aggregated to uphold privacy guarantees. Recent innovations in DP-SGD have primarily concentrated on ameliorating its computational and memory inefficiencies. TF-Privacy vectorizes the loss to calculate per-sample gradients through backpropagation, which is efficient in terms of memory but slow in execution Abadi et al. (2015). Opacus Yousefpour et al. (2021) and Rochette et al. (2019) enhance the training efficiency by employing the outer product method Goodfellow (2015), albeit at the cost of increased memory usage needed to store per-sample gradients. This memory overhead is mitigated in FastGradClip Lee & Kifer (2020) by distributing the space complexity across two stages of backpropagation, effectively doubling the time complexity. Additionally, ghost clipping techniques Goodfellow (2015), Li et al. (2021), Bu et al. (2022) allow for clipping per-sample gradients without full instantiation, optimizing both time and space, particularly when feature dimensions are constrained. Furthermore, Bu et al. (2023) introduces a 'book-keeping' (BK) method that achieves high throughput and memory efficiency while falling short in handling the long sequence lengths typical in LLM training.

While these methodologies have made significant strides in mitigating the extensive computational and memory demands typically associated with managing per-sample gradients in DP-SGD, they have not addressed the optimization of DP training from the perspective of GPU architecture and memory access. Additionally, the approaches detailed thus far do not cater effectively to the training of today's long-sequence LLMs. FlashDP aims to enhance the efficiency and feasibility of training

LLMs with long sequences under the constraints of differential privacy, ensuring both high performance and adherence to privacy standards.

**DP for Large Language Models.** The field of privacy-preserving LLMs is characterized by the use or exclusion of DP and its extensions. He et al. (2022) evaluated the precision equivalence of per-layer clipping with flat clipping on LLMs. Kerrigan et al. (2020) demonstrated that public pretraining could facilitate downstream DP fine-tuning, although they did not explore fine-tuning large pre-trained models using DP-SGD. Qu et al. (2021) explored the fine-tuning of BERT for language understanding tasks under local DP. Bommasani et al. (2021) suggested the potential for cost-effective private learning through fine-tuning large pre-trained language models. Anil et al. (2021) and Dupuy et al. (2022) extended these studies to BERT, pretraining and fine-tuning under global DP, respectively, with Anil et al. (2021) addressing datasets comprising hundreds of millions of examples, and Dupuy et al. (2022) reporting on datasets of utterances with relatively high $\epsilon$ values. Our research distinguishes itself by focusing on pre-training and fine-tuning large language models with high throughput and low memory usage.

## 3 UNDERSTANDING THE LIMITATIONS OF PREVIOUS METHODS

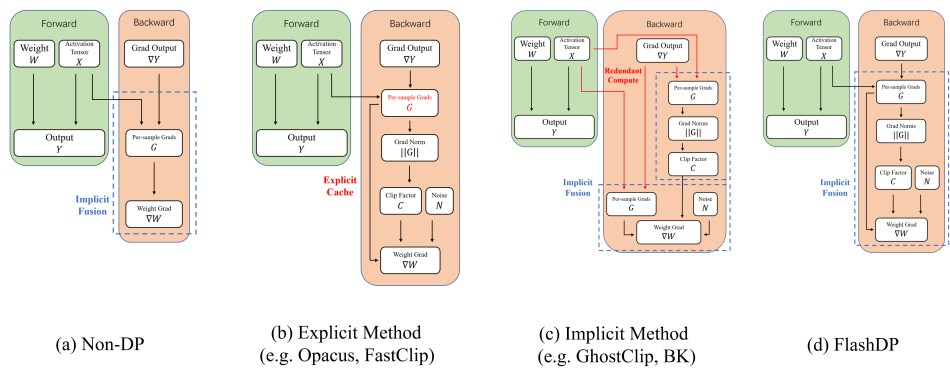

(a) Non-DP

(b) Explicit Method
(e.g. Opacus, FastClip)

(c) Implicit Method
(e.g. GhostClip, BK)

(d) FlashDP

Figure 2: Comparison of different training methods. (a) Non-DP: Basic training without DP. (b) Explicit Method (e.g., Opacus, FastClip): Stores per-sample gradients **G** (red explicit cache), increasing memory usage. (c) Implicit Method (e.g., GhostClip, BK): Reduces memory by recalculating gradients in fused manners (blue dotted box) but implicitly calculating the per-sample gradient twice, causing computational redundancy. (d) FlashDP: Optimizes gradient processing by consolidating computations into a single pass, reducing redundancy and memory use.

In this section, we introduce the previous non-DP, explicit, and implicate methods of DP-SGD from the GPU I/O perspective to see their weakness, which motivates our framework. Due to the space limit, please refer to Appendix A for the background on DP, Transformers, GPU architecture, and CUDA programming. As discussed in Section A.2, the linear operation is crucial in the architecture of LLMs, particularly within Multi-Head Attention (MHA) and Feedforward Network (FFN) modules. Given its significance, we utilize the linear operation as an exemplar to elucidate the training workflow on GPUs, as shown in Figure 2. See Appendix B for details.

In the standard non-private training workflow of a linear layer, the forward pass involves a matrix multiplication $Y = XW^{\mathsf{T}}$ between the activation tensor $X \in \mathbb{R}^{B \times T \times P}$ and the weight matrix $W \in \mathbb{R}^{D \times P}$, resulting in the output $Y \in \mathbb{R}^{B \times T \times D}$, where $B$, $T$, $P$, and $D$ denote the batch size, sequence length, input feature dimension, and output feature dimension, respectively. The backward pass calculates the output gradient $\nabla_Y \in \mathbb{R}^{B \times T \times D}$ and the weight gradient $\nabla_W \in \mathbb{R}^{D \times P}$ via $\nabla_W = \sum_B \sum_T (\nabla_Y)^{\mathsf{T}} X$. Figure 2 (a) illustrates this process, showing that the activation tensor $X$ and weights $W$ are stored in HBM for efficient access during computations, while intermediate operations utilize SRAM to enhance memory access time and throughput.

The explicit DP-SGD workflow, as depicted in Figure 2 (b), categorizes the process into four stages to ensure privacy adherence by explicitly managing per-sample gradients. **Stage 1** involves computing per-sample gradients $\mathbf{G} = \sum_T \nabla_Y^T X$ using batched GEMM operations on SRAM to minimize latency, with subsequent storage of the gradients back to HBM. **Stage 2** requires reloading these gra-

dients to compute their norm $\|\mathbf{G}\| = \sqrt{\sum_D \sum_P \mathbf{G}^2}$, then storing the results back in HBM. **Stage 3** includes loading the gradients and their norms for the per-layer clipping operations, ensuring that no gradient norm exceeds the predefined threshold $C$, with the clipped gradients $\mathbf{G}'$ written back to HBM. **Stage 4** focuses on adding Gaussian noise to the clipped gradients in SRAM for privacy preservation, followed by their aggregation for model updates, and storing the final noisy gradient $\nabla_W$ back in HBM. This explicit handling of per-sample gradients not only increases memory usage but also complicates processing due to frequent memory swaps and disrupts efficient GPU utilization by breaking down kernel fusion strategies, becoming notably impractical for LLMs with their extensive parameter and gradient sizes, severely impacting training efficiency.

The implicit DP-SGD workflow, illustrated in Figure 2 (c), employs a method such as GhostClip to recalculate gradients in a fused manner, thus circumventing the need for explicit storage of per-sample gradients. **Stage 1** consolidates the first three stages of the explicit method into a single fused computational step, where the activation tensor $X$ and output gradient tensor $\nabla_Y$ are loaded into SRAM. Per-sample gradient tensor $\mathbf{G}$ recalculations, norm calculations, and the per-layer clipping are integrated into one operation, minimizing latency and avoiding repeated data transfers to HBM. **Stage 2** mirrors the explicit method's final stage, where the recalculated and clipped gradients $\mathbf{G}'$ undergo Gaussian noise addition in SRAM, followed by aggregation and storage in HBM for model updates. This approach reduces memory usage but increases computational load due to the redundancy of multiple gradient recalculations, which can significantly extend training times, particularly for LLMs with extensive sequence lengths, rendering the method less practical due to the increased time complexity proportional to $T$.

To address the previous limitations, the subsequent section will introduce FlashDP, a novel strategy designed to address these inefficiencies by rethinking the execution pipeline of DP-SGD. Without delving into specifics here, FlashDP's architecture will streamline the integration of per-sample gradient computation and clipping, potentially reducing the operational bottlenecks observed in existing methods.

# 4 FLASHDP ALGORITHM DESIGN

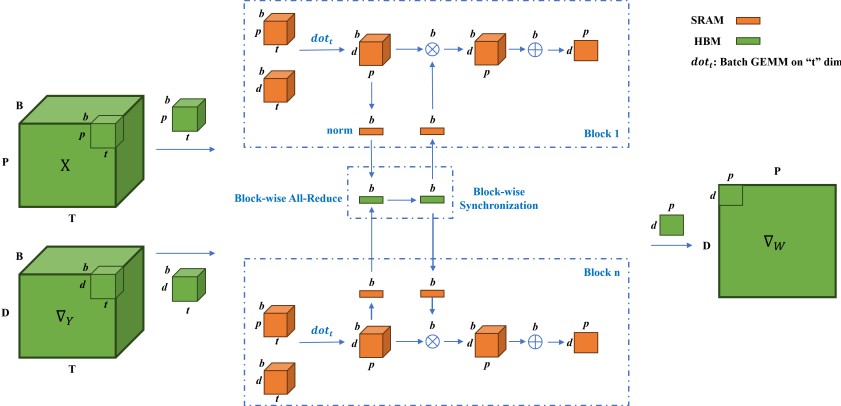

Figure 3: **Illustration of FlashDP.** It depicts the core algorithm design of FlashDP. Its features are integrated with on-chip per-sample gradient norm calculations. The workflow incorporates block-wise all-reduce and synchronization to facilitate efficient norm aggregation. SRAM (orange) and HBM (green) are optimally utilized to manage memory efficiently, addressing the kernel fusion challenges and reducing computational redundancy inherent in traditional DP-SGD implementations.

## 4.1 ALGORITHMIC ENHANCEMENTS IN FLASHDP

FlashDP introduces a suite of algorithmic enhancements designed to reconcile the computational demands and memory constraints associated with DP-SGD. At the heart of these enhancements is the Block-wise All-Reduce algorithm, which integrates several critical operations into a unified kernel execution, thereby optimizing on-chip memory utilization and enhancing computational throughput.

**Efficient Kernel Fusion through Block-wise All-Reduce.** Central to FlashDP's strategy is our proposed Hierarchical Reduction Architecture (HRA), which encompasses more than just reduction operations. HRA is a structured approach that manages the computation and synchronization of data across various stages, beginning with intra-block reduction of gradient norms within individual GPU blocks. This phase employs an HRA-based reduction strategy executed in shared memory, culminating in a single norm scaler per block. Such a design significantly reduces the data footprint necessary for subsequent inter-block communications, optimizing the efficiency of the all-reduce operation across the GPU grid.

Following the compact intra-block reduction, FlashDP coordinates a global all-reduce operation across blocks, which computes a global gradient norm crucial for consistent gradient clipping across the entire mini-batch. Efficiently handled in HBM thanks to the minimized data size from earlier reductions, this step avoids the common memory bottlenecks typically associated with large-scale data operations in HBM, thus maintaining high computational throughput.

---

**Algorithm 1** Algorithm: FlashDP with Block-wise All-Reduce on GPUs

---

**Require:** Input activation tensor $X \in \mathbb{R}^{B \times T \times P}$ and output gradient tensor $\nabla_Y \in \mathbb{R}^{B \times T \times D}$ in GPU HBM

**Require:** Clipping threshold $C$, noise scale $\sigma$

**Require:** Block dimensions $b$, $t$, $d$, and $p$ for batch size, sequence length, output features, and input features, respectively.

1: Split block for output gradient tensor $B_{\nabla_Y} \in \mathbb{R}^{b \times t \times d}$, input activation tensor $B_X \in \mathbb{R}^{b \times t \times p}$ based on GPU on-chip SRAM size $M$.

2: **for** each training backward iteration **do**

3:     **for** each block input index $i_p = 1, 2, \ldots, \frac{P}{p}$ in parallel **do**

4:         **for** each block output feature $i_d = 1, 2, \ldots, \frac{D}{d}$ in parallel **do**

5:             **for** each block batch size $i_b = 1, 2, \ldots, \frac{B}{b}$ in parallel **do**

6:                 Load output gradient block $B_{\nabla_Y}$ and input activation block $B_X$ from HBM to SRAM.

7:                 Compute per-sample gradients block $B_G = \sum_T B_{\nabla_Y}^T B_X$ on-chip SRAM.

8:                 Intra-block Reduce: Compute per-sample gradients norm square block $\|B_G\|^2 = \sum_d \sum_p B_{\mathbf{G}}^2$ on-chip SRAM.

9:                 Inter-block Reduce: Offload all per-sample gradients norm square blocks $\|B_G\|^2$ from SRAM to HBM, and perform block-wise all-reduce.

10:                 Block-wise synchronization: Wait until all blocks finish the all-reduce operation to get all-reduced per-sample gradients norm square blocks $\|B_G\|^{2'}$.

11:                 Upload $\|B_G\|^{2'}$ from HBM to SRAM.

12:                 Compute clipped per-sample gradients block $B'_G = B_G / \max\left(1, \frac{\sqrt{\|B_G\|^{2'}}}{C}\right)$ on-chip SRAM.

13:                 Add noise to clipped per-sample gradients block and aggregate to compute parameter gradient block $B_{\nabla_W} = \sum_b B'_G + \mathcal{N}(0, \sigma^2 C^2 \mathbf{I})$ on-chip SRAM.

14:                 Offload parameter gradient block $B_{\nabla_W}$ from SRAM to HBM.

15:             **end for**

16:         **end for**

17:     **end for**

18: **end for**

19: Return entire parameter gradient $\nabla_W$.

---

The strategic implementation of HRA not only facilitates these reductions but also orchestrates synchronized updates and data consistency across the GPU architecture. By managing data flow from the point of loading through to final computation and storage, HRA ensures that the most intensive computations are confined to the faster, on-chip memory. This methodical approach leverages the GPU's capabilities to facilitate high-performance differentially private training, minimizing memory and bandwidth overhead.

The practical implementation and operational dynamics of the FlashDP approach are thoroughly illustrated in Algorithm 1 and visually depicted in Figure 3. FlashDP innovatively reduces the four distinct stages typically involved in explicit DP-SGD into a **single streamlined stage**. This con-

solidation is achieved without adding any extra computational steps, thereby enhancing the overall efficiency of the process. Here is a detailed breakdown of this single streamlined stage:

**Optimized Block Processing and Memory Management (Line 1-6).** Initially, FlashDP partitions the input activation tensor $X$ and the output gradient tensor $\nabla_Y$ into blocks based on the SRAM capacity. This strategic partitioning is crucial for managing the limited on-chip memory more effectively and ensuring that data transfers between the HBM and SRAM are minimized.

**Fused Computation of Gradients and Norms (Line 7-8).** Within the GPU's SRAM, FlashDP simultaneously computes the per-sample gradients block and their norms square (intra-block reduce) for each block. This computation leverages the GPU's powerful batched GEMM operations, enabling it to handle large data sets efficiently.

**Block-wise All-Reduce (Line 9-11).** After computing the gradient norms, FlashDP performs a Block-wise All-Reduce operation in parallel to aggregate these norms across all blocks (inter-block reduce). This all-reduce operation is crucial for obtaining a global view of gradient norms square, which is necessary for consistent gradient clipping across the entire batch. This step is executed efficiently within the SRAM, reducing the latency and memory bandwidth requirements typically associated with inter-GPU communications.

**Gradient Clipping and Noise Addition in SRAM (Line 12-13).** Following the gradient and norm calculations, clipping is performed directly on the chip. Each gradient is scaled according to the computed norms and a predefined clipping threshold $C$, ensuring compliance with DP standards. Immediately after clipping, Gaussian noise based on the noise scale $\sigma$ and the clipping threshold is added to each gradient block.

**Efficient Parameter Aggregation (Line 14-19).** The final step in the FlashDP algorithm involves aggregating the noisy, clipped gradients across all blocks and batches directly within SRAM. This aggregation is optimized to minimize memory accesses, ensuring that only the final gradient used for the model update is transferred back to HBM.

## 4.2 ADAPTIVE KERNEL IMPLEMENTATION

The implementation of the FlashDP algorithm leverages the robust and versatile capabilities of the PyTorch framework Paszke et al. (2019), which is renowned for its intuitive handling of automatic differentiation and dynamic computational graphs. One of the critical features of our implementation involves customizing PyTorch's autograd functionality to accommodate the specific needs of differential privacy during the training of deep neural networks. To this end, operators that necessitate trainable parameters are intricately defined by wrapping them within PyTorch's autograd function.

However, implementing the Block-wise All-Reduce algorithm has presented unique challenges, primarily due to the limitations of CUDA's programming model in facilitating block-wise synchronization. Block-wise synchronization is essential in our algorithm; without it, clip operations might be executed prematurely, while the inter-block reduce operation is still incomplete, leading to numerical inaccuracies in the computation of per-sample gradients' norm squares. There are two primary methods to implement synchronization: 1. cooperative groups (CG) [1] and 2. adaptive kernel. We opted for the second method because the grid synchronization required by CG necessitates launching all blocks simultaneously, which is impractical for DP applications.

To address this limitation, FlashDP's implementation employs an adaptive approach. Instead of relying on a monolithic kernel to perform the entire Block-wise All-Reduce operation, the process is split across different kernels, which are executed iteratively over the batch dimension. This iterative approach allows for synchronization points between the execution of kernels, using the inherent block synchronization that occurs at kernel launch and completion.

The execution flow in FlashDP is as follows: (1) **Intra-block Reduction**: Each block computes the norms of its gradients and performs an HRA-based reduction within the shared memory. This step employs a shuffle-reduce mechanism, optimizing intra-block operations by minimizing memory footprint and synchronization overhead. This results in a single norm value per block. (2) **Inter-block Reduction**: Each block transfers the outcome of its intra-block reduction to the HBM. This transfer is facilitated through atomic operations for several reasons. Firstly, the result of the intra-

---

[1]https://developer.nvidia.com/blog/cooperative-groups/

block reduction comprises only a single element, and each block elects only one thread to perform the atomic operation on this element. This approach minimizes potential bottlenecks, as the differing execution speeds across blocks prevent serious serialization issues. Secondly, atomic operations benefit from acceleration by the hardware instruction set, ensuring that these operations are executed swiftly and efficiently. (3) **Inter-kernel Synchronization**: After the completion of the inter-block reduction, FlashDP leverages the termination of the kernel as a natural synchronization point. At this juncture, all blocks have finished their individual reductions. (4) **Iterative Kernel Launch**: For each batch element, a new kernel is launched serially, maintaining synchronization across kernels. This approach involves broadcasting operations where source operands are dimensionally disparate, ensuring uniform data handling across computational units.

This implementation strategy, while divergent from the ideal single-kernel solution, allows FlashDP to function effectively within the current constraints of CUDA. It underscores FlashDP's adaptability and represents a practical solution to the block synchronization challenge, ensuring accurate gradient norm calculations essential for maintaining the model's differential privacy. Our analysis on memory and access in Appendix C shows the utility of this implementation.

## 5 EXPERIMENTS

Our experimental suite is methodically designed to assess the robustness and efficiency of FlashDP across a range of training paradigms and hardware configurations. We explore FlashDP's performance in terms of memory efficiency and throughput under varying batch sizes, its adaptability to Automatic Mixed Precision (AMP) training (Appendix Section E.2), its consistency across different sequence lengths, and its scalability when employing Distributed Data Parallel (DDP) and Pipeline Parallel (PP) techniques.

Table 1: **Differential Batch-size Analysis.** The table displays a multi-panel comparison of memory usage and throughput for four differential privacy methods—NonDP, Opacus, GhostClip, BK, and FlashDP—across different batch sizes B (1, 2, 4, and 8) when applied to GPT-2 models of varying sizes (small, medium, and large). Instances of '-' in the table indicate scenarios where the corresponding method failed to execute due to memory constraints.

| Model | B | Memory Usage (MB x1e4) | | | | | Throughput (tokens/sec x1e4) | | | | |
|---|---|---|---|---|---|---|---|---|---|---|---|
| | | NonDP | Opacus | GhostClip | BK | FlashDP | NonDP | Opacus | GhostClip | BK | FlashDP |
| GPT2-small | 1 | 0.50 | 0.75(x1.50) | **0.46(x0.92)** | 0.53(x1.06) | 0.50(x1.00) | 2.84 | 0.91(x0.32) | 0.57(x0.20) | 1.56(x0.54) | **1.83(x0.64)** |
| GPT2-medium | | 1.26 | 1.53(x1.21) | **1.12(x0.89)** | 1.68(x1.33) | 1.26(x1.00) | 1.10 | 0.42(x0.38) | 0.39(x0.35) | 0.75(x0.68) | **0.86(x0.78)** |
| GPT2-large | | 2.48 | 3.99(x1.61) | **2.17(x0.88)** | 2.73(x1.18) | 2.48(x1.00) | 0.58 | 0.25(x0.43) | 0.27(x0.46) | 0.40(x0.69) | **0.51(x0.89)** |
| GPT2-small | 2 | 0.87 | 1.30(x1.49) | **0.79(x0.91)** | 1.01(x1.16) | 0.87(x1.00) | 3.22 | 1.68(x0.52) | 0.92(x0.29) | 1.91(x0.59) | **2.32(x0.72)** |
| GPT2-medium | | 2.07 | 2.89(x1.39) | **1.87(x0.90)** | 2.44(x1.18) | 2.07(x1.00) | 1.28 | 0.74(x0.58) | 0.59(x0.46) | 0.81(x0.63) | **1.02(x0.80)** |
| GPT2-large | | 3.91 | 4.79(x1.23) | **3.53(x0.90)** | 4.81(x1.23) | 3.91(x1.00) | 0.68 | 0.38(x0.56) | 0.38(x0.56) | 0.45(x0.66) | **0.59(x0.87)** |
| GPT2-small | 4 | 1.53 | 2.07(x1.35) | **1.44(x0.94)** | 1.68(x1.09) | 1.53(x1.00) | 3.60 | 2.42(x0.67) | 1.42(x0.39) | 2.24(x0.62) | **2.59(x0.72)** |
| GPT2-medium | | 3.58 | 4.26(x1.19) | **3.33(x0.93)** | 4.00(x1.12) | 3.58(x1.00) | 1.42 | 0.90(x0.63) | 0.81(x0.57) | 0.95(x0.67) | **1.13(x0.80)** |
| GPT2-large | | 6.60 | - | **6.15(x0.93)** | 6.60(x1.00) | 6.60(x1.00) | 0.76 | - | 0.50(x0.66) | 0.53(x0.70) | **0.64(x0.84)** |
| GPT2-small | 8 | 2.86 | 3.44(x1.20) | **2.72(x0.95)** | 2.86(x1.00) | 2.86(x1.00) | 3.80 | 2.64(x0.69) | 1.92(x0.51) | 2.40(x0.63) | **2.72(x0.72)** |
| GPT2-medium | | 6.60 | - | **6.24(x0.95)** | 6.60(x1.00) | 6.60(x1.00) | 1.52 | - | 0.99(x0.65) | 1.03(x0.68) | **1.19(x0.78)** |
| GPT2-large | | - | - | - | - | - | - | - | - | - | - |

### 5.1 EXPERIMENTAL SETUP

Our experiments utilize the Wikitext dataset Merity (2016) and are conducted on NVIDIA A100 (80GB) GPUs using the PyTorch framework Paszke et al. (2019). We assess the performance of FlashDP across various configurations by comparing it with established explicit methods Opacus Yousefpour et al. (2021), and implicit method GhostClip Li et al. (2021) and BK Bu et al. (2023), under different training paradigms. The tested models include GPT-2 Radford et al. (2019) with a sequence length of 1024 and the TinyLlama Zhang et al. (2024) and Llama Touvron et al. (2023) models, both with a sequence length of 2048. Our evaluations mainly focus on memory usage (MB) and throughput (tokens/sec) to determine the efficiency. We also show the loss of the validation data to measure the utility of private pre-training. Unless specified otherwise, the settings for each experiment use GPT-2 models with a sequence length of 1024, and Llama models with a sequence length of 2048, employing the AdamW optimizer as the base. More experimental settings can be found in Appendix D.

### 5.2 RESULTS OF BATCH SIZE & MICRO BATCH SIZE

Efficient batch processing is crucial in LLM training due to its high computational and memory demands. By examining both batch and micro-batch sizes, we assess FlashDP's ability to manage

memory more effectively and maintain high throughput. This also tests the practicality of gradient accumulation (GA), which allows larger effective batch sizes by splitting them into smaller, manageable micro-batches. The experiment results of different micro batch sizes can be seen in Appendix E.1.

In Table 1, FlashDP was benchmarked against traditional DP-SGD methods like Opacus, GhostClip, and BK, as well as a non-DP (NonDP) configuration, demonstrating superior memory efficiency and throughput. FlashDP utilized approximately 38% less memory than Opacus and nearly matched the NonDP configuration while processing the GPT-2 large model at a batch size of 1. It achieved a throughput nearly double that of Opacus and only slightly lower than NonDP, showcasing its effective balance between privacy preservation and computational efficiency. Opacus exhibited the highest memory usage, which escalated with batch size, leading to failure at a batch size of 8. GhostClip, while more memory-efficient than Opacus, suffered from reduced throughput at higher batch sizes due to gradient re-computation. BK's performance was intermediate, lacking distinct advantages. Overall, FlashDP not only maintained lower memory usage and higher throughput than the DP methods across all batch sizes but also approached the efficiency of NonDP configurations.

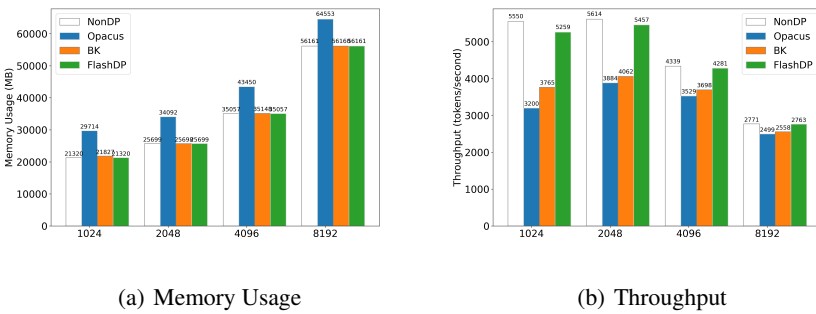

(a) Memory Usage  (b) Throughput

Figure 4: **Memory and Throughput Comparison for TinyLlama with Varied Sequence Lengths Using Flash Attention.** (a) Memory usage across sequence lengths of 1024, 2048, 4096, and 8192. (b) Throughput measured in tokens per second across the same sequence lengths.

### 5.3 RESULTS OF SEQUENCE LENGTH

In the training of LLMs, the ability to process long sequences of data is crucial for enhancing the model's capability to understand and generate coherent, contextually rich text.

**Memory Usage Analysis.** As illustrated in Figure 4 (a), there is a clear trend of increasing memory usage with longer sequence lengths across all methods, which is expected due to the larger computational requirements. However, FlashDP always maintains the same GPU memory usage as NonDP, especially at the highest sequence length of 8192. This indicates that FlashDP's method is particularly effective at managing the increased memory demands, thus facilitating the scalability of models trained with long sequences.

**Throughput Performance.** Figure 4 (b) highlights throughput in terms of tokens per second at varying sequence lengths. FlashDP consistently maintains higher throughput compared to Opacus and BK across all sequence lengths, with its performance closely approaching that of the NonDP method. This efficiency in throughput underlines FlashDP's capability to handle larger sequence lengths without significant compromises in processing speed, a critical factor for training usable and responsive LLMs.

The experimental data clearly demonstrates FlashDP's superior memory management and throughput efficiency across a range of sequence lengths. The ability of FlashDP to handle longer sequences with minimal increase in memory usage and only slight reductions in throughput is particularly impressive.

### 5.4 RESULTS OF DISTRIBUTED TRAINING

Distributed Data Parallel (DDP) Li et al. (2020) and Pipeline Parallel (PP) Kim et al. (2020) are two advanced techniques crucial for scaling the training of LLMs efficiently across multiple GPUs or nodes.

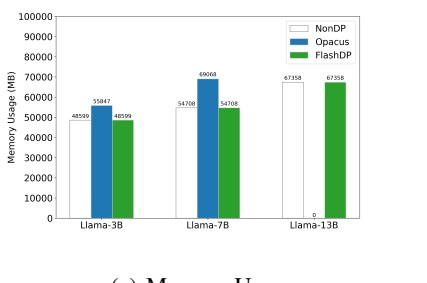
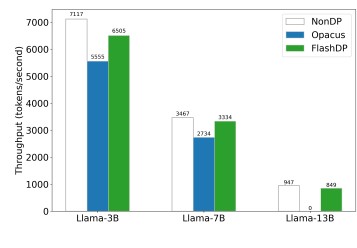

(a) Memory Usage           (b) Throughput

Figure 5: **Memory and Throughput for Llama Models Using Pipeline Parallel Training.** (a) Memory usage for Llama-3B, Llama-7B, and Llama-13B models. (b) Throughput in tokens per second across these model sizes. A value of 0 indicates out of memory.

**Distributed Data Parallel (DDP).** Figure 8 in Appendix illustrates the performance of different methods in a DDP setting across GPT-2 models of varying sizes. FlashDP showcases superior memory usage efficiency and higher throughput across all model sizes when compared to Opacus and BK. Notably, even as the model size increases, FlashDP maintains a competitive edge close to the NonDP benchmarks, highlighting its effective parameter distribution and gradient computation across multiple GPUs. This is crucial in scenarios where training speed and model scalability are priorities.

**Pipeline Parallel (PP).** In the PP scenario depicted in Figure 5, FlashDP was tested with Llama models varying from 3 billion to 13 billion parameters. The results indicate that FlashDP not only scales efficiently with increasing model size but also demonstrates significant throughput improvements compared to Opacus and BK. Particularly, FlashDP's ability to handle the largest model (Llama-13B) with minimal throughput degradation illustrates its robustness in managing extensive computational loads, characteristic of PP environments.

## 5.5 RESULTS OF UTILITY

Table 2: **FlashDP Pretrain Precision validation on GPT2-small with different privacy $\epsilon$.**

| Method | Validation loss | | |
|---|---|---|---|
| | $\epsilon = 0.2$ | $\epsilon = 0.5$ | $\epsilon = 0.8$ |
| DP-SGD | 4.8082 | 4.8063 | 4.8061 |
| FlashDP | 4.8082 | 4.8063 | 4.8061 |

In our study, FlashDP is meticulously optimized for DP-SGD, focusing on enhancing GPU I/O and system-level efficiencies without altering the fundamental algorithmic components of DP-SGD. We conducted experiments on utility with GPT-2 small to support this, whose results are shown in Table 2. From the table, we can easily see that FlashDP demonstrates an identical validation loss to that of standard DP-SGD across all privacy levels.

## 6 CONCLUSION

In this paper, we introduce FlashDP, a novel approach for integrating differentially private SGD (DP-SGD) into the training of large language models (LLMs) while enhancing memory efficiency and computational throughput. By optimizing GPU input/output operations, FlashDP significantly reduces the memory transaction overhead, allowing it to achieve near-non-private throughput levels while maintaining strict privacy standards. Central to FlashDP's strategy is the Block-wise All-Reduce algorithm, which integrates several critical operations into a unified kernel execution. To achieve this, we propose a Hierarchical Reduction Architecture (HRA), which encompasses more than just reduction operations. Moreover, we employ an adaptive kernel approach to implement HRA, which addresses the limitations of CUDA's programming model in facilitating block synchronization. Our experiments demonstrate that FlashDP reduces memory usage to levels comparable with non-private methods and increases throughput, making the efficient training of substantial models like the Llama 13B feasible on modern hardware. The minimal interference in the training process and the maintenance of computational precision suggest that FlashDP could significantly advance the adoption of DP in sectors where privacy is crucial, making secure and efficient machine learning more accessible for a wider range of applications.

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
