# A PRELIMINARIES

## A.1 DIFFERENTIAL PRIVACY

**Definition 1.** *(Differential Privacy Dwork et al. (2006)) Given a data universe $\mathcal{X}$, two datasets $X, X' \subseteq \mathcal{X}$ are adjacent if they differ by one data example. A randomized algorithm $\mathcal{M}$ is $(\varepsilon, \delta)$-differentially private if for all adjacent datasets $X$, $X'$ and for all events $S$ in the output space of $\mathcal{M}$, we have $\Pr(\mathcal{M}(X) \in S) \le e^\varepsilon \Pr(\mathcal{M}(X') \in S) + \delta$.*

**Differentially Private Stochastic Gradient Descent (DP-SGD) Abadi et al. (2016).** DP-SGD is an adaptation of this principle for machine learning models, where privacy is preserved during the training process by modifying the gradient computation.

In the context of a model parameterized by weights $\theta$ for loss $\mathcal{L}$, the standard SGD update is modified in DP-SGD to include a mechanism for privacy preservation. Specifically, the gradient $\nabla \mathcal{L}(\theta, x_i)$ for each training example $x_i$ is first computed, and then processed as follows to incorporate privacy:

1. **Clipping:** Each gradient is clipped to a maximum norm $C$, defined as: $g'_i = g_i \min(1, \frac{C}{\|g_i\|_2})$, where $g_i = \nabla \mathcal{L}(\theta, x_i)$.

2. **Noise Addition:** Gaussian noise is added to the aggregated clipped gradients to ensure differential privacy:

$$\tilde{g} = \frac{1}{B} \sum_{i=1}^{B} g'_i + \mathcal{N}(0, \sigma^2 C^2 I)$$

   where $B$ is the batch size, and $\sigma$ is the noise scale, determined by the privacy budget, subsampling rate, and iteration number.

The model parameters are then updated using the noisy, aggregated gradient: $\theta \leftarrow \theta - \eta \tilde{g}$, where $\eta$ is the learning rate. This approach to privacy-preserving training addresses the fundamental trade-off between accuracy and privacy by controlling the granularity of the updates through the parameters $C$ and $\sigma$.

In this work, we actually use Differentially Private Adam (DP-Adam) instead of DP-SGD. While DP-Adam incorporates the same mechanisms for gradient clipping and noise addition as described for DP-SGD, it also leverages the adaptive learning rates characteristic of Adam. The detailed algorithms can be found in Algorithm 2-4.

---

**Algorithm 2** Common Gradient Processing in DP-SGD and DP-Adam

---

**Require:** $\mathcal{L}(\theta, x_i)$: Loss function for parameter $\theta$ and input $x_i$
**Require:** $C$: Clipping threshold
**Require:** $\sigma$: Noise scale
**Require:** $B$: Batch size
1: **for** $i = 1$ to $B$ **do**
2:     Compute gradient: $g_i = \nabla \mathcal{L}(\theta, x_i)$
3:     Clip gradient: $g'_i = g_i \min(1, \frac{C}{\|g_i\|_2})$
4: **end for**
5: Aggregate clipped gradients and add Gaussian noise: $\tilde{g} = \frac{1}{B} \sum_{i=1}^{B} g'_i + \mathcal{N}(0, \sigma^2 C^2 I)$

---

**Algorithm 3** DP-SGD Specific Steps

---

**Require:** $\theta$: Model parameters
**Require:** $\eta$: Learning rate
1: **for** each training step **do**
2:     Perform common gradient processing as in Algorithm 2
3:     Update model parameters: $\theta \leftarrow \theta - \eta \tilde{g}$
4: **end for**

---

---

**Algorithm 4** DP-Adam Specific Steps

---

**Require:** $m, v$: Estimates of the first and second moments (initially 0)
 1: **for** each training step **do**
 2:     Perform common gradient processing as in Algorithm 2
 3:     Update moment estimates: $m \leftarrow \beta_1 m + (1 - \beta_1)\tilde{g}$
 4:     $v \leftarrow \beta_2 v + (1 - \beta_2)\tilde{g}^2$
 5:     Compute adaptive learning rate: $\hat{\eta} = \eta/(\sqrt{v} + \epsilon)$
 6:     Update parameters: $\theta \leftarrow \theta - \hat{\eta}m$
 7: **end for**

---

### A.2    TRANSFORMERS

The transformer architecture, proposed by Vaswani et al. Vaswani et al. (2017), is predicated on self-attention mechanisms that process input tokens in parallel, significantly improving the performance and training efficiency of sequence-to-sequence tasks. This architecture has become the backbone of LLMs.

In a transformer model, the input tensor $\mathbf{X}$ of size $B \times T \times P$ (since we are considering LLM, so we only focus on text data as the input), where $B$ is the batch size, $T$ is the sequence length (number of tokens), and $P$ is the embedding size of a token, undergoes a series of transformations through multi-head self-attention and feedforward neural network blocks. For each token in the sequence, the transformer computes a weighted sum of all tokens in the input, where the weights are determined through the self-attention mechanism.

**Multi-Head Attention (MHA).** The attention mechanism is primarily built upon linear transformations where the query $\mathbf{Q}$, key $\mathbf{K}$, and value $\mathbf{V}$ matrices are obtained as follows:

$$\mathbf{Q} = \mathbf{X}\mathbf{W}_Q, \quad \mathbf{K} = \mathbf{X}\mathbf{W}_K, \quad \mathbf{V} = \mathbf{X}\mathbf{W}_V \tag{1}$$

where $\mathbf{W}_Q$, $\mathbf{W}_K$, and $\mathbf{W}_V$ are the weight matrices that are subject to training.

**Feedforward Network (FFN).** The FFN in the transformer consists of two linear transformations with a ReLU activation in between:

$$\text{FFN}(\mathbf{x}) = \text{ReLU}(\mathbf{x}\mathbf{W}_1)\mathbf{W}_2 \tag{2}$$

Here, $\mathbf{W}_1$ and $\mathbf{W}_2$ are the weight matrices, all of which are trainable parameters of the linear layers within the FFN.

**Layer Normalization (LN).** LN is applied post-attention and FFN in each layer of the transformer. It normalizes the output of each neuron to have a mean of zero and a variance of one, which are then scaled and shifted by the trainable parameter vectors $\boldsymbol{\gamma}$ and $\boldsymbol{\beta}$, respectively:

$$\text{LayerNorm}(\mathbf{x}) = \boldsymbol{\gamma} \odot \left( \frac{\mathbf{x} - \mu}{\sqrt{\sigma^2 + \epsilon}} \right) + \boldsymbol{\beta} \tag{3}$$

where $\mu$ and $\sigma^2$ are the mean and variance calculated over the last dimension of the input tensor $\mathbf{x}$, $\epsilon$ is a small constant added for numerical stability, and $\odot$ denotes element-wise multiplication. The layer normalization parameters $\boldsymbol{\gamma}$ (scale) and $\boldsymbol{\beta}$ (shift) are learned to optimally scale and shift the normalized data.

The key trainable parameters in the transformer model are:

1. Weights of the WHA mechanism, including query $\mathbf{W}_Q$, key $\mathbf{W}_K$, and value $\mathbf{W}_V$ matrices, each of size $P \times P$.

2. Position-wise FFN weights $\mathbf{W}_1$ of size $P \times H$ and $\mathbf{W}_2$ of size $H \times P$, where $H$ is the hidden layer size.

3. LN parameters $\boldsymbol{\gamma}$ and $\boldsymbol{\beta}$, which are vectors of size $P$.

It is important to highlight that the bulk of the trainable parameters in the transformer model stems from MHA and FFN modules, both of which consist of linear transformations. These linear parameters are responsible for the vast majority of transformations within the transformer and significantly

contribute to its parameter count. In contrast, the trainable parameters in LN represent a relatively smaller portion of the model's total parameters. Therefore, we focus on the linear parameters gradient computation.

**DP-SGD for Training Transformers.** The process of adapting DP-SGD to transformers is formalized as follows: For each batch of input data $X$ and corresponding loss function $\mathcal{L}$, compute the per-sample gradients $\mathbf{G}_\theta$ for all trainable parameters $\theta = \{\mathbf{W}_Q, \mathbf{W}_K, \mathbf{W}_V, \mathbf{W}_1, \mathbf{W}_2, \boldsymbol{\gamma}, \boldsymbol{\beta}\}$:

$$\mathbf{G}_\theta = \nabla_\theta \mathcal{L}(\theta, X) \in \mathbb{R}^{B \times |\theta|}. \tag{4}$$

where $\nabla_\theta \mathcal{L}(\theta, X)$ denotes the computation of gradients of the loss with respect to the parameters $\theta$ for the batch $X$.

A.3 GPU ARCHITECTURE AND CUDA PROGRAMMING

High performance in deep learning, particularly in operations like General Matrix to Matrix Multiplication (GEMM), is largely attributable to the parallel processing power of modern Graphics Processing Units (GPUs). The architectural design of GPUs, with their numerous cores and hierarchical memory systems, is optimized for the parallel execution of operations, making them ideal for the matrix-intensive computations required in neural network training.

**GPU Architecture.** At the heart of GPU's computational efficiency are its Streaming Multiprocessors (SMs), which are essentially multiprocessor units that execute a large number of threads concurrently. Each SM is a powerhouse of performance, containing a set of processing cores and a block of on-chip memory, primarily Shared Random Access Memory (SRAM), which includes registers and shared memory. Shared memory, an ultra-fast SRAM, allows threads within the same block to exchange data without involving the slower global memory (HBM), thus acting as a crucial facilitator for matrix blocking.

**CUDA and GEMM.** The quintessential challenge in optimizing GEMM lies in the meticulous orchestration of data movement and computation, an endeavor where matrix blocking emerges as a pivotal strategy. Leveraging the robust architecture of GPUs and the sophisticated abstractions provided by CUDA (Compute Unified Device Architecture), matrix blocking transforms the theoretical prowess of parallel computation into a practical performance paradigm.

**Principles of Matrix Blocking.** Matrix blocking, also known as matrix tiling, is a technique ingeniously conceived to enhance data locality and parallelism. It systematically partitions extensive matrix operands into smaller, manageable sub-matrices or 'blocks' that can be independently dispatched to the GPU's SMs. The judicious use of shared memory within SMs for these blocks reduces the frequency and volume of global memory accesses, a common bottleneck due to its higher latency. Blocking is pivotal in minimizing the communication overhead between the slow global memory and the fast but limited on-chip shared memory. This stratagem leverages the temporal and spatial locality by reusing data within the fast-access memory hierarchies, significantly reducing the volume of data shuttled to and from the global memory, thereby enhancing the computational throughput.

**Mathematical Formalization of Blocking GEMM.** Consider the GEMM operation defined as $\mathbf{C} = \mathbf{A} \times \mathbf{B}$, where $\mathbf{A} \in \mathbb{R}^{m \times n}$, $\mathbf{B} \in \mathbb{R}^{n \times p}$, and the resultant matrix $\mathbf{C} \in \mathbb{R}^{m \times p}$. Blocking decomposes this operation into smaller, tractable computations over blocks such that:

$$\mathbf{C}_{ij} = \sum_{k=1}^{N} \mathbf{A}_{ik} \times \mathbf{B}_{kj}, \tag{5}$$

where $N$ is the number of blocks, and each $\mathbf{C}_{ij}$, $\mathbf{A}_{ik}$, and $\mathbf{B}_{kj}$ represents a sub-matrix or block within $\mathbf{C}$, $\mathbf{A}$, and $\mathbf{B}$, respectively. The indices $i$, $j$, and $k$ denote the specific block within the partitioned matrices.

The dimensions of each block are chosen based on the GPU's shared memory constraints and the size of the SMs' thread blocks, enabling optimal utilization of resources. These dimensions are represented as $B_m \times B_n$ for $\mathbf{A}_{ik}$ and $B_n \times B_p$ for $\mathbf{B}_{kj}$, leading to a block $\mathbf{B}_C$ in size of $B_m \times B_p$ for $\mathbf{C}_{ij}$. Hence, the computational paradigm shifts to:

$$\mathbf{B}_{C_{ij}} = \sum_{k=1}^{B_n} (\mathbf{B}_{A_{ik}} \times \mathbf{B}_{B_{kj}}), \tag{6}$$

where each multiplication within the summation is an independent block-level GEMM that can be executed in parallel.

# B    DETAILS OF TRAINING WORKFLOW

## B.1    NON-PRIVATE TRAINING WORKFLOW

In the standard training regime without privacy constraints, the linear forward operation takes an activation tensor $X \in \mathbb{R}^{B \times T \times P}$ and a weight matrix $W \in \mathbb{R}^{D \times P}$, producing an output $Y \in \mathbb{R}^{B \times T \times D}$ according to the matrix multiplication $Y = XW^{\mathsf{T}}$, where B, T, P, and D indicate the batch size, sequence length (token length), feature dimension of input activation tensor $X$, and feature dimension of output activation tensor $Y$, respectively.

During the backward pass, the gradient of the output with respect to the loss, denoted by $\nabla_Y \in \mathbb{R}^{B \times T \times D}$, is computed to be of the same dimensions as the output tensor $Y$. Subsequently, the gradient with respect to the weight matrix $W$, denoted by $\nabla_W \in \mathbb{R}^{D \times P}$, is obtained by summing the product of the transpose of the gradient tensor of each batch item and the corresponding input tensor, expressed as $\nabla_W = \sum_B \sum_T (\nabla_Y)^{\mathsf{T}} X$, where $\sum_B$ represents the summation along the dimension $B$ (similar for other notations).

Figure 2 (a) illustrates the computational workflow for the forward and backward pass of a linear operation within this conventional training framework. As shown in the figure, the activation tensor $X$ and the weights $W$ reside in HBM, which allows for rapid parallel access and is typically used for storing larger datasets and model parameters during GPU computations. The intermediate dot products and summations are handled using SRAM, shown in orange, which is faster than HBM and suitable for storing temporary, small blocks of data during computation. This setup minimizes memory access time and maximizes throughput.

## B.2    EXPLICIT DP-SGD WORKFLOW

Figure 2 (b) terms the explicit method (e.g., Opacus, FastClip), demonstrates the traditional DP approach where per-sample gradients are stored explicitly, resulting in increased memory usage due to the retention of individual gradient information for noise addition and clipping. The explicit DP-SGD workflow is normally organized into four distinct stages to ensure adherence to privacy constraints:

**Stage 1: Per-sample Gradient Computation.** At this initial stage, the activation tensor $X \in \mathbb{R}^{B \times T \times P}$ and the output gradient tensor $\nabla_Y \in \mathbb{R}^{B \times T \times D}$ are loaded in blocks from the HBM to the on-chip SRAM. The per-sample gradients tensor $\mathbf{G} \in \mathbb{R}^{B \times D \times P}$ is computed by performing the operation $\mathbf{G} = \sum_T \nabla_Y^T X$ directly on the SRAM to minimize latency, effectively implementing a batched GEMM operation, where each slice of $\mathbf{G}$ is per-sample gradient. After computation, the per-sample gradients are written back to the HBM for further processing.

**Stage 2: Gradient Norm Computation.** The computed per-sample gradients $\mathbf{G}$ are again loaded into SRAM in smaller blocks. The norm of per-sample gradient is then computed on-chip, $\|\mathbf{G}\| = \sqrt{\sum_D \sum_P \mathbf{G}} \in \mathbb{R}^B$. Then, this norm calculation is stored in HBM.

**Stage 3: Gradient Clipping.** This stage involves loading both the per-sample gradients $\mathbf{G}$ and its norm $\|\mathbf{G}\|$ from the HBM into SRAM. The clipping operation is performed by computing $\mathbf{G}' = \mathbf{G}/\max\left(1, \frac{\|\mathbf{G}\|}{C}\right)$ (this division occurs in dimension B), ensuring that each gradient's norm does not exceed the clipping threshold $C$. The clipped gradients $\mathbf{G}'$ are then stored back in HBM.

**Stage 4: Noise Addition and Aggregation.** In the final stage, the clipped per-sample gradients $\mathbf{G}'$ are loaded into SRAM, and Gaussian noise $\mathcal{N}(0, \sigma^2 C^2 \mathbf{I})$ is added to each, according to the specified noise scale $\sigma$. This process ensures differential privacy by obfuscating the contributions of individual training examples. The noisy, aggregated gradient for the weight update, $\nabla_W = \sum_B \mathbf{G}' + \mathcal{N}(0, \sigma^2 C^2 \mathbf{I})$, is computed and then written to HBM, ready for updating the model parameters.

**Limitations.** Standard DP-SGD requires the explicit storage of per-sample gradients in HBM, which is crucial for computing the gradient norms needed for clipping. This requirement substantially increases the memory footprint. This method becomes impractical for LLMs, which have large model parameters and gradients due to extended sequence lengths. The extensive memory needed to store these gradients often exceeds the available HBM capacity, leading to frequent data swapping between memory and processing units, which severely slows down the training process. Crucially, the computation of gradient norms breaks down standard kernel fusion strategies, preventing the efficient integration of gradient computation and subsequent processing steps into a single operation, resulting in increased latency and inefficient GPU utilization.

### B.3 IMPLICIT DP-SGD WORKFLOW

Figure 2 (c) illustrates the implicit method (e.g., GhostClip, BK), which optimizes the DP-SGD process by recalculating gradients in a fused manner, thereby avoiding the explicit storage of per-sample gradients. This approach reduces memory demands but introduces computational redundancy due to multiple gradient recalculations. The implicit DP-SGD workflow is normally organized into two distinct stages:

**Stage 1: Fused Computation (corresponds to Stage 1-3 of the explicit method).** In the implicit method, stages 1 through 3 of the explicit method are executed in a fused computational process. This involves loading the activation tensor $X \in \mathbb{R}^{B \times T \times P}$ and the output gradient tensor $\nabla_Y \in \mathbb{R}^{B \times T \times D}$ into SRAM. The per-sample gradients tensor $\mathbf{G} \in \mathbb{R}^{B \times D \times P}$ is recalculated by integrating gradient computation, norm calculation, and clipping into a single pass. This minimizes latency and avoids repeated data transfers to HBM. During this fused operation, the per-sample gradient norms are calculated $\|\mathbf{G}\|$ directly on the chip. Clipping is simultaneously performed by scaling the gradients: $\mathbf{G}' = \mathbf{G}/\max\left(1, \frac{\|\mathbf{G}\|}{C}\right)$, where $C$ is the clipping threshold. These operations are performed without storing the intermediate states, reducing the memory footprint.

**Stage 2: Noise Addition and Aggregation (corresponds to stage 4 of the explicit method).** The clipped gradients $\mathbf{G}'$ are recalculated and loaded into SRAM where Gaussian noise $\mathcal{N}(0, \sigma^2 C^2 \mathbf{I})$ is added, adhering to the specified noise scale $\sigma$. The final aggregate gradient is then computed and written back to HBM for the model update.

**Limitations of Implicit methods:** Implicit methods attempt to mitigate the high memory usage by segmenting the gradient computation and clipping it into several smaller, manageable tasks. However, these methods involve multiple recalculations of the per-sample gradients, which is computationally expensive. Specifically, for LLM training where the sequence length dimension $T$ is very large, the redundant computation required by these methods can lead to a significant increase in training time. The time complexity for per-sample gradient recalculations is $O(T)$ when $T$ is very large, which makes such methods impractically slow for pre-training LLMs.

## C ANALYSIS OF HBM MEMORY USAGE AND ACCESSES IN FLASHDP

The foundational design of FlashDP incorporates significant advancements in minimizing both HBM usage and accesses. This dual optimization plays a pivotal role in enhancing the computational efficiency and scalability of DP training for LLMs. Reducing HBM usage is crucial because it directly impacts the GPU's ability to manage large datasets and complex computations without exhausting available memory resources. Theorem C.1 illustrates that FlashDP's HBM memory usage is much lower than DP-SGD and almost equal to Non-private training.

**Theorem C.1.** *Let $B$, $T$, $P$, $D$ be the batch, token length, input, and output dimension, and let $X \in \mathbb{R}^{B \times T \times P}$, $\nabla_Y \in \mathbb{R}^{B \times T \times D}$, and $\nabla_W \in \mathbb{R}^{D \times P}$ be input, gradient output, and the gradient of weight in the linear. Non-Private training backward requires $BT(P + D) + DP$ (HBM) memory usage, DP-SGD requires $BT(P+D)+PD(B+1)+B$, and the FlashDP requires $BT(P+D)+DP+b$, where $b$ is the block size for $B$.*

Simultaneously, FlashDP drastically lowers the number of HBM accesses required during training. Each access to HBM, whether for reading or writing data, incurs a latency penalty. By minimizing these accesses, FlashDP alleviates bandwidth bottlenecks that can degrade the training performance. This is achieved by strategically leveraging faster, on-chip memory for the majority of the

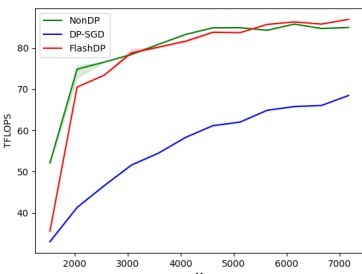

Figure 6: Comparative throughput of Non-private Training, DP-SGD, and FlashDP. This figure plots the throughput in TFLOPS against a unified model parameter size denoted by 'M', where M represents the equality of the dimensions P, D, and T. All experiments are implemented with batch-size = 16 and run on A100 (80GB).

data-intensive computations, significantly reducing the need to fetch or store data in HBM. This efficient memory usage prevents frequent swapping of data to and from slower storage media, thus maintaining high computational speeds and reducing the potential for memory overflow issues. In other words, FlashDP's throughput is much higher than that of DP-SGD and is almost equal to that of non-private training.

**Theorem C.2.** *With the same notations as in Theorem C.1, and let $M$ be the number of CUDA block, the Non-Private training backward requires $BT(P + D) + DP$ (HBM accesses), DP-SGD requires $BT(P + D) + PD(4B + 1) + B$, and FlashDP requires $BT(P + D) + DP + 2Mb$.*

**Comparative analysis.** In our comparative analysis of throughput performance, depicted in Figure 6, we observe the scaling behavior of FlashDP relative to NonDP and DP-SGD as a function of increasing model parameter size 'M' (we set $M = P = D = T$). The results reveal that as 'M' increases, FlashDP's throughput demonstrates a marked improvement over DP-SGD, suggesting that our approach significantly mitigates the performance degradation typically associated with DP. Notably, FlashDP's curve closely aligns with that of NonDP training, which underscores the efficacy of FlashDP in maintaining high throughput. This near convergence with NonDP throughput is particularly evident when 'M' exceeds 3000, highlighting FlashDP's scalability and potential for practical application in large-scale LLM training where differential privacy is a requirement. These findings corroborate our proposition that FlashDP's IO-grained approach to gradient clipping within GPU memory is not only innovative but also practically advantageous.

### C.1 PROOFS OF THEOREMS ON HBM MEMORY USAGE AND ACCESSES

This section provides detailed proofs for the Theorem C.1 and C.2 presented in the main text concerning the High Bandwidth Memory (HBM) usage and accesses for different training methodologies of large language models. These theorems compare the efficiency of Non-Private training, DP-SGD, and FlashDP, our proposed method.

*Proof.* In Non-Private training, the memory requirement includes the input tensor $X \in \mathbb{R}^{B \times T \times P}$, the gradient output tensor $\nabla_Y \in \mathbb{R}^{B \times T \times D}$, and the gradient of weights $\nabla_W \in \mathbb{R}^{D \times P}$. Thus, the total HBM usage is calculated as $BT(P + D) + DP$.

For DP-SGD, additional memory is required to store per-sample gradients $\mathbf{G} \in \mathbb{R}^{B \times D \times P}$ and per-sample gradients norm $\|\mathbf{G}\| \in \mathbb{R}^B$ for whole sample, contributing to a significant increase in memory usage, as reflected in the formula.. Thus, the total HBM usage is calculated as $BT(P + D) + DP(B + 1) + B$.

FlashDP optimizes memory by using the same base structure as Non-Private training but includes an additional minimal term for the norm block $B_{\|\mathbf{G}\|} \in \mathbb{R}^b$, significantly reducing the memory usage compared to DP-SGD. The total HBM usage is calculated as $BT(P + D) + DP + b$. □

*Proof.* In Non-Private training, the required accesses include loading and storing the input tensor $X \in \mathbb{R}^{B \times T \times P}$, the gradient output tensor $\nabla_Y \in \mathbb{R}^{B \times T \times D}$, and the gradient of weights $\nabla_W \in \mathbb{R}^{D \times P}$. The $X$ and $\nabla_Y$ should be uploaded and the $\nabla_W$ should be stored, leading to total HBM accesses of $BT(P + D) + DP$.

For DP-SGD, the tensor accesses increase significantly due to the handling of per-sample gradients $\mathbf{G} \in \mathbb{R}^{B \times D \times P}$ and per-sample gradients norms $\|\mathbf{G}\| \in \mathbb{R}^B$. Each per-sample gradient and norm must be individually loaded for computation and then stored back. This includes not only their initial computation but also the additional loads and stores for each gradient during the clipping and noise addition stages, resulting in total accesses of $BT(P + D) + PD(4B + 1) + B$, reflecting multiple reads and writes per sample.

FlashDP, leveraging the same base structure as Non-Private training, reduces HBM accesses by avoiding per-sample operations. It introduces efficient on-chip processing for the norm calculations and clipping in blocks, significantly reducing the need for frequent tensor movements. The primary memory movements in FlashDP are associated with loading the partitions of $X$, $\nabla_Y$, and storing $\nabla_W$, along with minimal additional accesses for the norm blocks $B_{\|\mathbf{G}\|} \in \mathbb{R}^b$ with $2M$ times. The total HBM accesses are thereby $BT(P + D) + DP + 2Mb$, representing a significant reduction compared to DP-SGD. □

## D ADDITIONAL EXPERIMENTS SETTINGS

**Batch Size & Micro Batch Size**   For the batch size experiment, we vary the batch sizes at 1, 2, 4, and 8, using GPT-2 models of small, medium, and large scales to test the method's scalability and efficiency. Similarly, in the micro-batch size experiment, we set the micro-batch sizes at 1, 2, 4, and 8, with a gradient accumulation step of 4.

**Experiments on Testing Utility**   We conduct an experiment to evaluate the performance of the GPT2-small model trained from scratch using DP-SGD and FlashDP under differential privacy constraints, with epsilon values set at 0.2, 0.5, and 0.8. The model is trained on the Fineweb-edu (Lozhkov et al., 2024) dataset. Key hyperparameters include a total batch size of 524,288 tokens, a micro batch size per device of 32, and a sequence length of 1024. We use a maximum learning rate of $6 \times 10^{-4}$ and a minimum learning rate of $6 \times 10^{-5}$, with weight decay set at 0.1 and gradient clipping at 1.0. The model undergoes training with a validation frequency every 250 steps and model saving every 5000 steps, using both DP-SGD and FlashDP, enabling differential privacy with delta set at $1 \times 10^{-5}$ and a clipping threshold of 100. The training aims to compare utility across different privacy levels and analyze the trade-offs between privacy and utility. We use the validation loss as the evaluation metric in Table 2.

**Sequence Length**   Long sequence lengths allow LLMs to maintain a broader context, crucial for tasks such as document summarization, question answering, and natural language understanding over extended dialogues. However, accommodating these longer sequences inherently increases the computational complexity and memory demands, particularly for gradient calculations during training. This experiment is designed to evaluate the efficiency of FlashDP in handling varying sequence lengths, specifically to address the challenge of increased memory usage associated with longer sequences in differential privacy settings. By training the TinyLlama model variant with Flash Attention Dao et al. (2022) at different sequence lengths, we aim to demonstrate FlashDP's capacity for memory management and throughput efficiency across these conditions. Such an analysis is essential to verify FlashDP's suitability for practical deployment in scenarios where deep contextual understanding is required.

In this experimental setup, we measure memory usage and throughput while training with sequence lengths of 1024, 2048, 4096, and 8192, at a fixed batch size of 1, comparing the performance of FlashDP against NonDP, Opacus, and BK methods. It is important to note that GhostClip does not support the Llama Model.

**Distributed Training**   DDP involves distributing the model's parameters across several devices, and each device computes gradients for a subset of the data independently. This method is beneficial for managing models that fit within the memory limits of a single GPU but need faster processing

through parallel execution. On the other hand, Pipeline Parallel (PP) splits the model's layers across different devices, allowing different parts of the model to be processed simultaneously. PP is particularly useful for very large models that exceed the memory capacity of individual GPUs, enabling concurrent processing of different stages of the model across the pipeline. The experiments with DDP and PP are designed to evaluate the effectiveness of FlashDP in a distributed training context, assessing its performance in terms of memory usage and throughput across various model sizes and batch sizes. These experiments are critical to demonstrate that FlashDP can maintain its efficiency and scalability when applied to state-of-the-art LLMs, which require substantial computational resources and sophisticated training mechanisms to manage their size and complexity.

In this setup, we explore the scaling capabilities of FlashDP using DDP on four A100 GPUs (80GB each) by training GPT-2 models of small, medium, and large sizes with fixed sequence lengths of 1024 and varying batch sizes of 8, 4, and 2. Additionally, PP experiments are conducted on Llama models of sizes 3B, 7B, and 13B to evaluate throughput and memory efficiency across different stages of the model pipeline. It is important to note that GhostClip and BK do not support the distributed modes we used.

# E  MORE EXPERIMENTAL RESULTS

## E.1  RESULTS OF MICRO BATCH SIZE

Table 3: **Micro Batch Size Analysis.** Comparing memory and throughput at varying micro batch sizes B (1, 2, 4, 8) and the same gradient accumulation steps (4) for GPT-2 sizes with differential privacy methods under consistent settings with Table 1.

| Model | B | Memory Usage (MB x1e4) | | | | | Throughput (tokens/sec x1e4) | | | | |
|---|---|---|---|---|---|---|---|---|---|---|---|
| | | NonDP | Opacus | GhostClip | BK | FlashDP | NonDP | Opacus | GhostClip | BK | FlashDP |
| GPT2-small | 1 | 0.51 | 0.97(x1.90) | 0.51(x1.00) | 0.71(x1.39) | **0.51(x1.00)** | 3.07 | 1.20(x0.39) | 0.60(x0.20) | 1.75(x0.57) | **1.86(x0.61)** |
| GPT2-medium | 1 | 1.26 | 1.69(x1.34) | **1.25(x0.99)** | 1.81(x1.44) | 1.26(x1.00) | 1.27 | 0.61(x0.48) | 0.45(x0.35) | 0.86(x0.68) | **0.91(x0.72)** |
| GPT2-large | 1 | 2.48 | 3.64(x1.47) | **2.46(x0.99)** | 3.21(x1.29) | 2.48(x1.00) | 0.67 | 0.39(x0.43) | 0.32(x0.46) | 0.47(x0.69) | **0.53(x0.89)** |
| GPT2-small | 2 | 0.87 | 1.15(x1.32) | 1.00(x1.15) | 1.06(x1.22) | **0.87(x1.00)** | 3.22 | 1.68(x0.52) | 0.92(x0.29) | 1.91(x0.59) | **2.32(x0.72)** |
| GPT2-medium | 2 | 2.07 | 2.88(x1.39) | **2.01(x0.97)** | 2.62(x1.27) | 2.07(x1.00) | 1.38 | 0.88(x0.64) | 0.65(x0.47) | 0.88(x0.64) | **1.04(x0.75)** |
| GPT2-large | 2 | 3.91 | 6.07(x1.55) | **3.83(x0.98)** | 4.43(x1.13) | 3.91(x1.00) | 0.74 | 0.46(x0.62) | 0.43(0.58) | 0.49(x0.66) | **0.59(x0.80)** |
| GPT2-small | 4 | 1.53 | 2.10(x1.37) | **1.48(x0.97)** | 1.73(x1.13) | 1.53(x1.00) | 3.72 | 2.49(x0.67) | 1.50(x0.40) | 2.30(x0.62) | **2.59(x0.70)** |
| GPT2-medium | 4 | 3.58 | 5.51(x1.54) | **3.46(x0.97)** | 4.04(x1.13) | 3.58(x1.00) | 1.48 | 0.97(x0.66) | 0.86(x0.58) | 0.99(x0.67) | **1.29(x0.87)** |
| GPT2-large | 4 | 6.60 | - | **6.45(x0.98)** | - | 6.60(x1.00) | 0.79 | - | 0.53(x0.67) | - | **0.65(x0.82)** |
| GPT2-small | 8 | 2.86 | 4.00(x1.40) | **2.78(x0.97)** | 3.06(x1.07) | 2.86(x1.00) | 3.87 | 2.60(x0.67) | 1.99(x0.51) | 2.44(x0.63) | **2.73(x0.71)** |
| GPT2-medium | 8 | 6.60 | - | **6.37(x0.97)** | 7.16(x1.08) | 6.60(x1.00) | 1.55 | - | 1.03(x0.66) | 1.05(x0.68) | **1.19(x0.77)** |
| GPT2-large | 8 | - | - | - | - | - | - | - | - | - | - |

Table 3 further explores the impact of varying micro batch sizes, a crucial factor for managing memory in constrained environments and optimizing the use of gradient accumulation steps. FlashDP consistently displayed minimal memory footprint increases and maintained high throughput efficiency, even as micro batch sizes increased. For example, at a micro batch size of 8 for the GPT-2 medium model, FlashDP's memory usage was $6.49 \times 10^4$ MB—marginally higher than its usage at smaller micro batch sizes and significantly lower than Opacus at the same size. This robust performance underscores FlashDP's effective management of memory, which is essential for scaling up the training of large models without excessive hardware requirements.

To be specific, 1) Opacus showed a consistent increase in memory usage as micro batch sizes increased, which is indicative of its inefficient memory handling under fragmented gradient computations. 2) GhostClip, while better in memory usage compared to Opacus, didn't scale as well in throughput, which decreased noticeably with larger micro batches, reflecting the computational cost of gradient recalculations. 3) BK displayed trends similar to Opacus but generally used slightly less memory and provided slightly better throughput, suggesting a more optimized handling of gradient accumulation steps. 4) FlashDP maintained minimal increases in memory usage with increasing micro batch sizes and consistently provided the highest throughput, highlighting its effective integration of operations within the computational workflow. To summarize, as the micro batch size increases, FlashDP's memory usage increases only slightly and still maintains the highest throughput, demonstrating its efficient memory management techniques.

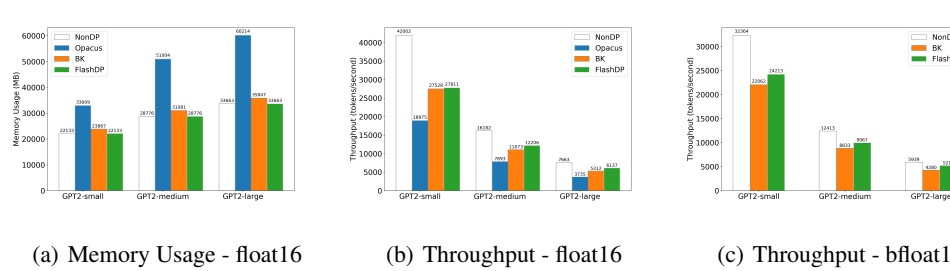

(a) Memory Usage - float16        (b) Throughput - float16        (c) Throughput - bfloat16

Figure 7: **Memory and Throughput Analysis of GPT-2 Models Using Automatic Mixed Precision (AMP) Training Across Float16 and BFloat16 Precision.**: (a) Demonstrates the memory usage for GPT-2 small, medium, and large models with Float16 precision. (b) shows throughput using Float16 precision, and (c) shows throughput with BFloat16 precision.

### E.2    RESULTS OF AMP TRAINING SCALABILITY

Automatic Mixed Precision (AMP) Micikevicius et al. (2017) training involves utilizing lower precision formats like float16 and bfloat16 within a training session to reduce computational demands and memory usage. This strategy is particularly valuable for large language models (LLMs), which typically require substantial computational resources. By employing AMP, training processes can be accelerated, and larger models or batches can be managed more efficiently without proportional increases in hardware capacity. The integration of differential privacy with AMP, especially in techniques like FlashDP, is critical for exploring the practical limits of DP-SGD. This experiment assesses how FlashDP adapts to AMP settings compared to other methods, and evaluates the impact on memory efficiency and processing speed, which are crucial for the scalability of private training in constrained environments.

In our experiments, we analyze GPT-2 models of varying sizes using batch sizes of 8, 4, and 2 across float16 and bfloat16 precision formats to measure memory usage and throughput, examining FlashDP's performance relative to NonDP, Opacus, and BK methods. It is important to note that GhostClip does not support AMP, and Opacus does not support the bfloat16 precision format.

**Memory Usage Analysis.** As depicted in Figure 7 (a), the memory usage across GPT-2 models of different sizes indicates that FlashDP, when utilizing AMP in both float16 and bfloat16 formats, maintains lower memory consumption compared to Opacus and BK, and closely approximates the NonDP configuration. This showcases FlashDP's effective use of AMP to minimize memory overhead, facilitating the training of large models under stringent privacy constraints.

**Throughput Analysis with Float16 and BFloat16.** In terms of throughput, Figure 7 (b) and 5(c) present a comprehensive look at the advantages of using float16 and bfloat16 precision formats under AMP. FlashDP consistently outperforms Opacus and BK in throughput metrics across both precision types. This is especially notable in larger model configurations, where the differences in throughput become more pronounced, highlighting FlashDP's capability to handle extensive computational loads efficiently. As demonstrated in Figure 5(b), FlashDP exhibits significant throughput advantages over the other DP methods. This performance is indicative of the efficient computational optimizations that FlashDP leverages within the AMP framework. As shown in Figure 7 (c), while bfloat16 typically offers slightly lower computational throughput than float16 due to its numerical properties, FlashDP's implementation still ensures that it outperforms other differential privacy methods. This underscores FlashDP's robust performance across varying precision settings.

## F    ADDITIONAL TABLES AND MORE FIGURES

Table 4: Comparison of Backward Propagation Methods.

| Method | Per-sample Gradient | | Implicit Fusion |
| --- | --- | --- | --- |
| | Cache | Recalculation | |
| Non-DP | ✗ | ✗ | ✓ |
| Explicit-DP | ✓ | ✗ | ✗ |
| Implicit-DP | ✗ | ✓ | ✓ |
| FlashDP | ✗ | ✗ | ✓ |

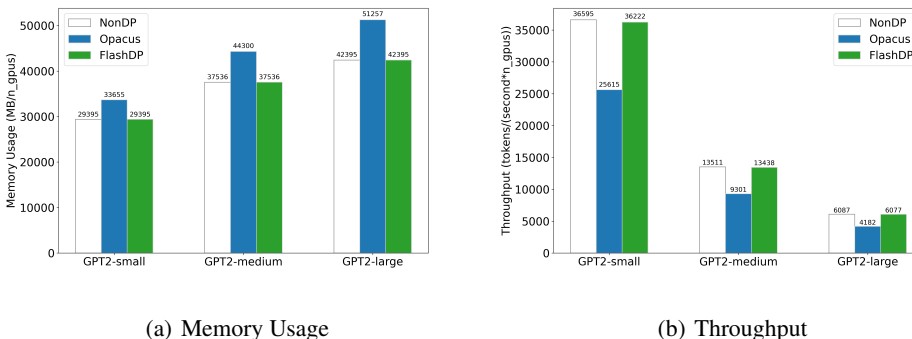

(a) Memory Usage  (b) Throughput

Figure 8: **Memory and Throughput for GPT Models Using Distributed Data Parallel Training.** (a) Memory usage for GPT-samll, GPT-medium, and GPT-large models. (b) Throughput in tokens per second across these model sizes. A value of 0 indicates out of memory.