# OpenReview forum: "FlashDP: Memory-Efficient and High-Throughput DP-SGD Training for Large Language Models"
_ICLR.cc/2025/Conference — ICLR 2025 Conference Withdrawn Submission_

### Official Review · Reviewer_5JeC · 2024-11-03

**Soundness:** 3
**Presentation:** 4
**Contribution:** 4
**Rating:** 5
**Confidence:** 2

**Summary:**

The paper considers the problem of efficiently computing per-sample clipped gradients in DP training. Clipping causes DP training to be more expensive than non-private training because e.g. computing batch gradients might be more efficient than computing single gradients and averaging them, or because we might need to reload parts of the gradient into memory to rescale them. The authors propose FlashDP, which uses implicit fusion of a wider number of steps of the DP batch gradient computation than past work. e.g. compared to the popular GhostClip, which recomputes per-sample gradients after computing the clip norm, FlashDP combines the gradient computation / norm computation / rescaling operation into a single pass. The authors conduct experiments to demonstrate the improvements of FlashDP over past work. Their experiments show e.g. higher gradient throughput than the Opacus library's standard clipping operation while using less memory, and also much higher throughput than GhostClip while using slightly more memory.

**Strengths:**

The main strength is the empirical comparisons which are quite favorable - it seems based on the results in this paper that most PyTorch users who are training with DP would want to use FlashDP if it were made widely available. Beyond that, a lot of care is taken in the presentation, e.g. there are plenty of nice figures, comparisons to past works are broken down carefully, the main hardware algorithm is broken down line-by-line, and the paper includes explanations for DP experts who might not be as familiar with e.g. transformer architecture or hardware details of GPU training. The technical contributions also seem pretty significant - the question of how to do clipping while computing gradients in a blockwise manner is highly nontrivial, since a priori it seems one needs to do a 'multi-pass' algorithm to learn the norm before clipping.

**Weaknesses:**

I think the paper has limited weakness, with the main potential weakness being that the empirical comparisons to past methods seem to consider only PyTorch implementations. However, even e.g. the original Opacus paper shows (in e.g. Appendix E of https://arxiv.org/pdf/2109.12298) that e.g. JAX implementations of DP-SGD may be more scalable than Opacus (e.g. in their figure 4.d, JAX pays a large up-front compilation cost, its per-iteration runtime is much smaller than Opacus). It would be interesting to understand how FlashDP compares to JAX DP-SGD implementations / if the techniques in FlashDP are restricted to training with PyTorch. However, since PyTorch is quite popular for both private and non-private training, and it would likely be a large amount of work to ask the authors to investigate this, I think this weakness is limited. Anyway, the comparison of FlashDP to non-private PyTorch training is quite favorable and suggests FlashDP compares favorably to non-PyTorch methods as well.

**Questions:**

Are the losses in Table 2 exactly the same? i.e., does this table show that the input -> output function computed by FlashDP and other implementations are exactly the same given the same noise, or are there possibly some differences in later digits (e.g. due to slight algorithmic differences or numerical stability issues) that the authors view as insignificant compared to the efficiency benefits?

---

> ### Author Response · Authors · 2024-11-24
> **Response by Authors**
>
> Thank you for your thorough review and the positive remarks on our work.
>
> For the weaknesses you mentioned, regarding the use of PyTorch, we opted for this framework due to its widespread adoption in both private and non-private training contexts. While our implementation is currently PyTorch-based, the core modifications we made to the CUDA code, which underlies both PyTorch and JAX, suggest that our method could indeed be extended to JAX. Adapting FlashDP to JAX would entail substantial work, including rewriting the model and submodule code for JAX and adjusting the underlying source code. We recognize the potential benefits of such an adaptation and plan to explore this in future work.
>
> As for your question about the losses reported in Table 2, they are indeed the same. FlashDP is designed to optimize the efficiency of GPU utilization without altering the fundamental computational processes of DP-SGD.

---

> ### Comment · Reviewer_5JeC · 2024-11-26
> **Thanks for response**
>
> Thanks to the authors for their response. While there is no further need to address the concerns in my initial review, reviewer 1eKM has raised some concerns about the presentation surrounding whether the current method vs. past work applies to per-layer vs flat gradients which I am in agreement with. In particular, I was under the impression FlashDP used flat clipping which made the comparisons to BK and GhostClip much more impressive. Now, as the other reviewers have pointed out, it is unclear how much of the improvement in the comparisons is because of the different end goals of flat clipping vs per-layer clipping, and how much is because of any novelties in FlashDP.
>
> For now I am reducing my score in light of the concerns raised in reviewer 1eKM's review and the resulting discussion. I would be happy to continue discussing in the followups to reviewer 1eKM's review to keep the discussion in one place, and happy to revisit my score if the discussion there can alleviate the reviewers' shared concerns. I apologize for the misunderstanding in my initial review and for reducing the score late in the review process.

---

### Official Review · Reviewer_UZDe · 2024-11-03

**Soundness:** 3
**Presentation:** 2
**Contribution:** 3
**Rating:** 6
**Confidence:** 1

**Summary:**

The vanilla DP-SGD has the memory and computational efficiency issue. Later-on methods try to improve this efficiency but still has additional cost in computations to save memory compared with non-private SGD. This paper proposes a method that encodes the per-sample clipping in the backward process. As evaluated in the experiment, when training with large language models, the proposed method matches the memory efficiency of  the non-private SGD and perform the best in terms of throughput compared with all implementations of DP-SGD.

**Strengths:**

1. The problem studied by this paper is important. The memory and computational cost introduced by DP-SGD is the bottleneck for applying it to large foundation models nowadays.
2. The empirical results on large language model look promising. The proposed method has a certain benefit over existing method: same memory cost as non-private SGD and 70-80% speed of non-private SGD.

**Weaknesses:**

1. The intuition and the benefit of the proposed method are not straightforward to me, although it can be because I am not familiar with the low-level design in the GPU and CUDA (my confidence is 1). It seems that in the algorithm, per-sample gradients for all weights still need to be saved at the same time, before the block-wise synchronization block. Opacus also explicitly save per-sample gradients. Why is the proposed method more memory-efficient than Opacus?
2. The details of the proposed method are not clear to me. Does the Algorithm 1 only describe the MLP layer? What is the details of other computations in the transformer?
3. Some definitions are not well-introduced in the main paper; for example the abbreviation GEMM and HBM are directly mentioned in main paper without the full short phrase. Again, it is likely because i am not familiar with the area, but it would be better to have more description so the audience can be wider.

**Questions:**

Please check the weakness section. Moreover,
1. Can the proposed method be applied to other architecture such as convolutional neural network in computer vision?
2. Is it possible that the method can be implemented as a package s.t. it can be easily used for other variants of transformer?
3. What's the reason that the memory usage of ghostclip is even more efficient than NonDP?

---

> ### Author Response · Authors · 2024-11-24
> **Response by Authors**
>
> Thank you for your insightful comments and questions. We appreciate your acknowledgment of the importance of our work and its potential benefits. Below, we address your points to clarify the aspects of our method:
>
> ## For the weaknesses you mentioned:
>
> 1. We explain the background on GPUs, CUDA, and chunked matrix multiplication in detail in Appendix Section A.3. Let's explain it here in layman's terms: the memory in a GPU can be roughly divided into two parts, the large but relatively slow High Bandwidth Memory (HBM), and the small but fast Static Random Access Memory (SRAM), which we usually refer to as the GPU memory. We can think of the per-sample gradient as an intermediate variable in the actual gradient computation, and Opacus stores this complete intermediate variable in the HBM, while we store this intermediate variable in chunks in SRAM when we compute the matrix multiplication in chunks. In this way, we both reduce the utilization of the HBM and increase the speed.
>
> 2. Our algorithm does not apply only to the MLP layer, but to the more basic Linear layer. almost all of the parameters in the transformer come from the Linear layer, including the attention as well as the MLP. this is explained in detail in the original Appendix Section A.2.
>
> 3. Technical Terminologies: The GEMM (General Matrix Multiply) and HBM (High Bandwidth Memory) are usually used in systems areas. Due to page constraints, we have explained these terms in the original Appendix Section A.3.
>
> ## For the questions you mentioned:
>
> 1. For the convolution operation, although it is not discussed in this paper due to space constraints, it can be easily transformed into a GEMM operation by a simple transformation. [[1](https://inria.hal.science/inria-00112631/document)] and [[2](https://arxiv.org/abs/1704.04428)]
>
> [1] https://inria.hal.science/inria-00112631/document
> [2] https://arxiv.org/abs/1704.04428
>
> 2. We have implemented it as a package and provided our complete code and detailed documentation in an anonymous repository, accessible here: [FlashDP4Review](https://anonymous.4open.science/r/FlashDP4Review-5CBE/).
>
> 3. The observed difference in memory usage between GhostClip and NonDP implementations with PyTorch can primarily be attributed to the underlying memory management mechanisms. GhostClip implements a strategy where it explicitly deletes the accumulated gradient to conserve memory, which can be observed in their [code implementation](https://github.com/lxuechen/private-transformers/blob/18ccc4eab7355e4ac96051a82434796f6aa4624b/private_transformers/privacy_engine.py#L503). This approach contrasts with NonDP (PyTorch), which employs a memory pool strategy to optimize allocation time during runtime. In the NonDP setup, used memory, such as the accumulated gradient, is placed into a memory pool. This means that once a computation requires more memory, PyTorch allocates it from the memory pool instead of performing new allocations. This approach speeds up memory allocation but can lead to slightly higher memory usage than what is strictly necessary for the computation at hand, as documented in the [PyTorch documentation](https://pytorch.org/docs/stable/notes/cuda.html).

---

> > ### Comment · Reviewer_UZDe · 2024-11-26
> >
> > Thanks for your response. "Weaknesses" in the review was actually asking why Algorithm 1 is doing the gradient clipping with only the linear layer and it is independent of other layers. There was no "per-layer gradient clipping" described in the submission. After carefully reading the review of Reviewer 1Ekm, the response to that review, and the revision during the rebuttal, I think now i understand more for this detail and I raise similar concern to Reviewer 1Ekm: Gholst-clipping and opacus are actually doing full-gradient clipping, and the algorithm here is doing per-layer gradient clipping.
> >
> > Here are my additional concern and question:
> > 1. In your response for Reviewer 1Ekm of "This indicates that their clipping is confined to individual layers, aligning with the concept of per-layer clipping", this is actually not true. They just calculate the per-layer norm one by one and the norm for the full gradient is calculated later on for the full-gradient clipping.
> > 2. It is not clear to me that the method proposed in this paper is really benefitting from the per-layer gradient or the advanced memory-level usage, because per-layer gradient itself can save memory: similar to ghost-clipping, in their first backward when calculating the per-layer norm for a layer, they can at the same time calculate the clipped gradient for this layer and release the per-sample gradient for this layer. I think your method essentially does the same thing, but with a potentially better memory-efficient implementation. I am curious how much memory saving in your method is exactly from the memory-efficient implementation.

---

### Official Review · Reviewer_1Ekm · 2024-11-04

**Soundness:** 2
**Presentation:** 3
**Contribution:** 2
**Rating:** 3
**Confidence:** 3

**Summary:**

The authors propose FlashDP — an algorithm for efficient gradient computation in DP-SGD training. It is based on Hierarchical Reduction Architecture, which implements a multistage scheme, beginning with intra-block All-Reduce and then proceeding to inter-block reductions. To implement this block-wise All-Reduce, the authors used an adaptive kernel approach. The proposed method was empirically evaluated for both speed and memory efficiency using GPT-2, TinyLLaMA, and LLaMA models.

**Strengths:**

1. The differential private pre-training of large language models represents a significant research challenge, and this paper makes a valuable contribution toward advancing this direction.

2. The authors conducted comprehensive experimental evaluations, comparing their method against both non-private baselines and alternative DP-SGD implementations in different experimental setups.

**Weaknesses:**

Major weaknesses:

1. The main limitation of this work is the absence of complete implementation for the proposed algorithm (let alone the code for experiment reproduction) or at least detailed pseudocode. The description provided in Algorithm 1 is insufficient as it remains too high-level - for instance, it lacks crucial implementation details of the Block-wise All-Reduce operation, which, as the authors themselves acknowledge (line 356), represents a key technical challenge. For engineering-focused papers presenting complex algorithmic implementations, I consider the availability of source code to be essential, making this limitation particularly **crucial**.

2. A fundamental challenge in DP-SGD implementation, as I understand it, stems from the requirement to compute **the full gradient norm across all layers** for per-example clipping (for example, this issue is thoroughly discussed in Section 4 of the prior work [1]). This requirement has led previous algorithms to either compute gradients twice (first to calculate the gradient norm across all layers) or store gradients for all layers in a buffer (to enable clipping without recomputation once the total norm is determined). However, the current work notably lacks any discussion of this crucial aspect (the substring "layer" appears only in the supplementary material and in different contexts). Without addressing this fundamental consideration, it becomes challenging to understand how the proposed implementation achieves compliance with the DP-SGD algorithm while avoiding additional gradient recomputation and memory overhead.

Minor weaknesses:

1. The authors omitted the GhostClip method from Figure 4 and both GhostClip and BK methods from Figure 5, without providing any justification for these omissions. This appears particularly problematic given that these methods were substantially more competitive with FlashDP than the retained Opacus baseline (according to Table 3, GhostClip even outperformed FlashDP in terms of memory usage).

2. Lines 8-10 of the Algorithm 1 describe an All-Reduce operation between blocks for gradient norm computation. Such an All-Reduce operation necessitates the presence of multiple blocks that have computed values to reduce. However, according to the pseudocode in Algorithm 1, all *for* loops are executed sequentially. The authors should provide clarification regarding this algorithmic detail.

3. Since the official LLaMA release did not include a 3B-sized model, I assume that the authors used a custom LLaMA-like model with 3B parameters. If so, the authors should clearly state this and provide a complete description of this model's architecture.

[1] Li et. al., Large language models can be strong differentially private learners. ICLR 2022.

**Questions:**

See Weaknesses.

---

> ### Author Response · Authors · 2024-11-24
> **Response by Authors**
>
> Thank you for your valuable feedback and detailed review. We appreciate the opportunity to clarify and improve our submission based on your insights.
>
> ## For the major weaknesses you mentioned:
>
> 1. Code Availability for Reproducibility: We acknowledge the critical importance of reproducibility in research. To address this, we have provided our complete implementation and detailed documentation in an anonymous repository, accessible here: [FlashDP4Review](https://anonymous.4open.science/r/FlashDP4Review-5CBE/).
>
> 2. Gradient Clipping Methodology: In our implementation, we adopt per-layer clipping, which is increasingly recognized as a standard practice in differential privacy for large language models (LLMs). This approach has been shown to effectively match or even surpass the state-of-the-art in various language and vision tasks, as documented in the following studies: [[1](https://arxiv.org/abs/2110.05679)], [[2](https://arxiv.org/abs/2210.00038)], [[3](https://arxiv.org/abs/2206.07136)]. These references demonstrate that per-layer clipping can be seamlessly integrated into existing codebases with minimal modifications. Importantly, this method achieves compliance with differential privacy guarantees while maintaining the performance benefits of full gradient clipping but with significantly reduced memory overhead and computational complexity. [[4](https://arxiv.org/abs/2212.01539)] Recent papers also have shown that per-layer clipping could have comparable performance with the original full gradient clipping.
>
> [1] https://arxiv.org/abs/2110.05679
> [2] https://arxiv.org/abs/2210.00038
> [3] https://arxiv.org/abs/2206.07136
> [4] https://arxiv.org/abs/2212.01539
>
> ## For the minor weaknesses you mentioned:
>
> 1. Exclusion of Methods in Figures: The absence of GhostClip and BK methods from certain figures was due to their incompatibility with the specific experimental setups tested. For example, in Figure 4, the GhostClip method does not support the Llama model structure, and in Figure 5, the BK method does not support pipeline parallelism. Please refer to the documentation [private-transformers](https://github.com/lxuechen/private-transformers) and [fast-differential-privacy](https://github.com/awslabs/fast-differential-privacy) for more details. We have clarified these exclusions and ensured detailed justifications in Appendix Section D of the revised version.
>
> 2. Algorithm Execution Details: The "for loops" in Algorithm 1 (lines 3, 4, and 5) are designed to execute in parallel, not sequentially. We have revised the paper.
>
> 3. Model Configuration: We used a model configuration consistent with OpenLlama's 3B model, as detailed here: [OpenLlama-3B](https://huggingface.co/openlm-research/open_llama_3b).

---

> ### Comment · Reviewer_1Ekm · 2024-11-25
> **Follow-up concerns**
>
> Thank you for your response. Some of my concerns have been addressed. However, I would like to raise some remaining major concerns:
>
> **Regarding per-layer clipping:**
>
> >we adopt per-layer clipping, which is increasingly recognized as a standard practice in differential privacy for large language models (LLMs). This approach has been shown to effectively match or even surpass the state-of-the-art in various language and vision tasks, as documented in the following studies: [[1](https://arxiv.org/abs/2110.05679)], [[2](https://arxiv.org/abs/2210.00038)], [[3](https://arxiv.org/abs/2206.07136)].
>
> I have carefully re-examined papers [1,2] specifically looking for experimental comparisons or at least mentions of per-layer clipping, but I could not find any such references. Moreover, as I mentioned in my initial review, these algorithms seem to be specifically designed to address the challenges of global flat clipping ([1] performs backward pass twice, while [2] additionally stores gradients with respect to the outputs of the all linear layers, and all these overheads just to compute global per-example gradient norm). To substantiate the claim that "per-layer clipping is documented in the following studies," I would kindly request the authors to explicitly point out the specific pages and paragraphs in these papers where per-layer clipping is discussed.
>
> > [[4](https://arxiv.org/abs/2212.01539)] Recent papers also have shown that per-layer clipping could have comparable performance with the original full gradient clipping.
>
> While this work indeed addresses per-layer clipping and appears significantly more relevant to the authors' problem formulation than [1,2], I notice that the authors neither compare their approach with the implementation from [4] nor mention this work in their paper. I would appreciate if the authors could explain why this relevant research is not referenced in their study.
>
> To summarize, I have a major concern regarding the validity and fairness of the experimental setup. Specifically, as I understand:
>
> * The authors' **main baselines** are methods from [1,2], which were **developed and designed to effectively solve a different problem** (flat clipping instead of per-layer clipping).
>
> * Meanwhile, other effective algorithms, such as [4], which are specifically **designed for efficient implementation of per-layer clipping** and would provide a more appropriate comparison, are **not included in the paper**.
>
> Also, in my opinion, Differentially Private optimizers with flat clipping and with per-layer clipping represent distinct approaches. Therefore, the fact that the authors specifically focus on per-layer clipping should be explicitly stated in the paper to avoid potential misinterpretation.
>
> **Regarding absence of GhostClip and BK methods from certain figures**
>
> I would again kindly request that the authors explicitly point to the specific location in the documentation of these frameworks where the mentioned limitations are stated. For instance, I could not find any substring "llama" in the README of repository [5].
>
> [1] [https://arxiv.org/abs/2110.05679](https://arxiv.org/abs/2110.05679)
>
> [2] [https://arxiv.org/abs/2210.00038](https://arxiv.org/abs/2210.00038)
>
> [3] [https://arxiv.org/abs/2206.07136](https://arxiv.org/abs/2206.07136)
>
> [4] [https://arxiv.org/abs/2212.01539](https://arxiv.org/abs/2212.01539)
>
> [5] [private-transformers](https://github.com/lxuechen/private-transformers)

---

> > ### Author Response · Authors · 2024-11-26
> > **Response to Your Follow-up Concerns (Part 1)**
> >
> > Thank you for your detailed follow-up.
> >
> > > I have carefully re-examined papers [1,2] specifically looking for experimental comparisons or at least mentions of per-layer clipping, but I could not find any such references. Moreover, as I mentioned in my initial review, these algorithms seem to be specifically designed to address the challenges of global flat clipping ([1] performs backward pass twice, while [2] additionally stores gradients with respect to the outputs of the all linear layers, and all these overheads just to compute global per-example gradient norm). To substantiate the claim that "per-layer clipping is documented in the following studies," I would kindly request the authors to explicitly point out the specific pages and paragraphs in these papers where per-layer clipping is discussed.
> >
> > We would like to highlight that per-layer clipping is a standard practice in efficiently training large models with Differential Privacy (DP). Consequently, many papers discussing DP efficiency may not explicitly mention the use of per-layer clipping. We want to clarify where these practices are implicitly addressed in the literature:
> >
> > 1. GhostClip [1]: Refer to **Section 4.2 (their core algorithm) Paragraph 2, 3, and 4** of the GhostClip paper. Here, the discussion focuses on operations at the level of **each linear layer**. Although the term "per-layer" is not used explicitly, the algorithm's description demonstrates per-layer operations. To quote: "Let $\mathbf{a} \in \mathbb{R}^{B \times T \times d}$ be the input to **a linear layer** with weight matrix $\mathbf{W} \in \mathbb{R}^{p \times d}$, and $\mathbf{s} \in \mathbb{R}^{B \times T \times p}$ be the output with $s_{i,j} = \mathbf{W}a_{i,j}$. ...". This indicates that their clipping is confined to individual layers, aligning with the concept of per-layer clipping.
> >
> > 2. BK [2], In **Algorithm 1 (their core algorithm) Line 8 and 9**, BK describes their gradient processing with explicit reference to individual layers, illustrating a layer-by-layer approach to gradient clipping: \
> > "for layer $ l \in \{L, L-1, \ldots, 1\} $ do " \
> > "Compute sum of clipped gradients $ G_l = a^{(l)T} \text{diag}(C_1, C_2, \ldots) \frac{\partial \mathcal{L}}{\partial \mathbf{s}_{(l)}} $" \
> > This description clearly points to a per-layer clipping of gradients, consistent with per-layer clipping practices.

---

> > ### Author Response · Authors · 2024-11-26
> > **Response to Your Follow-up Concerns (Part 2)**
> >
> > > While this work [4] indeed addresses per-layer clipping and appears significantly more relevant to the authors' problem formulation than [1,2], I notice that the authors neither compare their approach with the implementation from [4] nor mention this work in their paper. I would appreciate if the authors could explain why this relevant research is not referenced in their study.
> >
> > Our selection of baselines, including Opacus, GhostClip, and BK, was driven by their specific contributions to **enhancing memory usage and throughput** in DP-SGD implementations, which align closely with the core objectives of our research.
> > 1. Opacus is widely recognized as a foundational DP-SGD framework, providing a baseline for naive DP-SGD implementations.
> > 2. GhostClip represents an advance over Opacus by optimizing memory usage through an innovative clipping technique.
> > 3. BK is the state-of-the-art baseline. It improves computational throughput and maintains almost the same memory usage compared with GhostClip.
> >
> > In contrast, the primary contribution of work [4] is in evaluating the equivalence of per-layer clipping with flat clipping in terms of **precision**. While this is undoubtedly important, it diverges from the main goals of our study, which concentrate on performance metrics rather than precision equivalence. Therefore, while work [4] offers valuable insights into clipping methodologies, it did not align directly with the specific improvements we aimed to explore and achieve in our DP-SGD implementation. Also, given that our comparisons already include the current state-of-the-art (BK), adding more baselines might not necessarily enhance the clarity or value of our experimental validations.
> >
> > However, we recognize the importance of work [4] and agree that citing this significant research in our related work section is necessary.
> >
> > > Also, in my opinion, Differentially Private optimizers with flat clipping and with per-layer clipping represent distinct approaches. Therefore, the fact that the authors specifically focus on per-layer clipping should be explicitly stated in the paper to avoid potential misinterpretation.
> >
> > In fact, your reminder helped us recognize that although many of the papers mentioned above use per-layer clipping by default in their DP training without explicitly stating so, this practice can confuse readers. We have decided to explicitly note this in our revised version to prevent any confusion.
> >
> > > I would again kindly request that the authors explicitly point to the specific location in the documentation of these frameworks where the mentioned limitations are stated. For instance, I could not find any substring "llama" in the README of repository [5].
> >
> > Please refer to the section titled "Currently supported Hugging Face models" in the README of the repository [5], where you will notice the absence of support for the Llama model. Regarding the distributed setup discussed in our text, while repositories [5] and [6] do not explicitly state their limitations on supportability, our empirical observations from real-world executions confirm the constraints we mentioned.
> >
> > [1] https://arxiv.org/abs/2110.05679 \
> > [2] https://arxiv.org/abs/2210.00038 \
> > [3] https://arxiv.org/abs/2206.07136 \
> > [4] https://arxiv.org/abs/2212.01539 \
> > [5] [private-transformers](https://github.com/lxuechen/private-transformers) \
> > [6] [fast-differential-privacy](https://github.com/awslabs/fast-differential-privacy)

---

> > > ### Comment · Reviewer_5JeC · 2024-11-26
> > > **Chiming in**
> > >
> > > I'm not the original reviewer, but wanted to comment on these pointers by the author:
> > >
> > > > GhostClip [1]: Refer to Section 4.2 (their core algorithm) Paragraph 2, 3, and 4 of the GhostClip paper. Here, the discussion focuses on operations at the level of each linear layer. Although the term "per-layer" is not used explicitly, the algorithm's description demonstrates per-layer operations.
> > >
> > > They say preceding Section 4.2:
> > >
> > > > Overall, the per-example gradient norm of any network without parameter sharing can be computed in a layer-by-layer fashion
> > > with only one per-example gradient tensor for a single layer being instantiated at any time.
> > >
> > > Hence my understanding of this paper is that they are computing the entire per-example gradient norm in a layer-by-layer fashion, and the procedure they are describing here is one step in the layer-by-layer procedure, not that the intent is to compute a per-layer norm for the purpose of per-layer clipping.
> > >
> > > >BK [2], In Algorithm 1 (their core algorithm) Line 8 and 9...
> > >
> > > In the same Algorithm, lines 6 and 7 compute the norm of the whole gradient by summing the squared norms of the layers and compute the clipping factor using the norm, I think it is clear that their algorithm is for clipping the whole gradient rather than per-layer gradient.

---

> ### Comment · Reviewer_UZDe · 2024-11-26
>
> I'm also not the original reviewer, and I would like to move my thread to here as well for better context. After reading this thread and the revised text, I now understand that the method is doing per-layer gradient. I have related responses and concerns:
> 1. "This indicates that their clipping is confined to individual layers, aligning with the concept of per-layer clipping" is actually not true. They just calculate the per-layer norm one by one and the norm for the full gradient is calculated later on for the full-gradient clipping.
> 2. It is not clear to me that the method proposed in this paper is really benefitting from the per-layer gradient or the advanced memory-level usage, because per-layer gradient itself can save memory: similar to ghost-clipping, in their first backward when calculating the per-layer norm for a layer, they can at the same time calculate the clipped gradient for this layer and release the per-sample gradient for this layer. Or in the equivalent words, it seems not fair to compare the proposed algorithm with the full-gradient clipping methods, and given the current comparison in the paper, it is unclear how much memory saving is from the advanced memory-level implementation.

---

> ### Comment · Reviewer_1Ekm · 2024-11-26
> **Re: Follow-up concerns**
>
> I thank the authors for their detailed response. Their specific references to the relevant sections in articles [1,2] have confirmed that we are indeed addressing the same material in our discussion.
>
> The key question from which all my current major concerns stem is the following: "Are the implementations of FlashDP, GhostClip[1], and BK[2] mathematically equivalent?" To avoid potential misunderstandings, I would like to present my current understanding of the situation systematically and in detail.
>
> 1. **There are two different clipping strategies** in the field of DP optimization: 1. Flat clipping and 2. Group-wise clipping. The difference between these two strategies is described in [3], Section 2, page 3, paragraphs 3, 4. Specifically, in the case of Flat clipping, there exists a **single group** of parameters $\theta$ and threshold $C$, while for Group-wise clipping, the parameters are divided into **$K$ disjoint sets** {$\theta_k$}$_{k=1}^K$, each with its own clipping threshold $C_k$.
> 2. **These two strategies are not mathematically equivalent** when $K > 1$. By this, I mean that for any value $C>0$ for Flat clipping and any values $C_k>0, k=1,\dots,K,$ there exists a vector $g$ such that Flat clipping and Group-wise clipping will return different results $g_{\text{flat}} \neq g_{\text{group}}.$
> 3. **Per-layer clipping is a special case of Group-wise clipping**, where $K$ equals the number of layers $L$ and each set $\theta_k$ is some instance of torch.nn.Parameter $W_l$ (or its equivalent in another deep learning framework).
> 4. **FlashDP supports only per-layer clipping**.
> 5. **GhostClip (Section 4 in [1]) and BK (Algorithm 1 in [2]) implement Flat clipping.** I draw this conclusion based on the following considerations: For GhostClip, from Section 4.1 paragraph 2 in [1]  where the authors describe how to calculate $\|\nabla \mathcal{L}_i\|_2$, i.e., the **full gradient norm**, using per-layer gradient norms $\{\|\nabla_W^{(1)}\mathcal{L}_i\|_2,\dots, \|\nabla_W^{(L)}\mathcal{L}_i\|_2\}$ and also from Algorithm 2 in Appendix C in [2] (page 15); For BK, from lines 3-7 of Algorithm 1 in [2], where they compute the value of $C_i$, which requires knowledge of $\|\frac{\partial \mathcal{L}_i}{\partial W}\|_F^2$, i.e., the **full gradient norm**.
>
> If the sequence of statements above is correct, then it follows that the FlashDP algorithm is not mathematically equivalent to GhostClip and BK.
>
> If this is the case, I return to the concerns I mentioned above. Specifically, I think that:
> 1. since FlashDP and its baselines implement different mathematical functions, the experimental comparisons are not fair and do not allow for drawing conclusions about the efficiency of FlashDP.
> 2. the effectiveness of FlashDP should be verified through comparisons with other implementations of per-layer clipping, such as the per-layer clipping from [3] (see Figure 1).
>
> [1] https://arxiv.org/abs/2110.05679
>
> [2] https://arxiv.org/abs/2210.00038
>
> [3] https://arxiv.org/abs/2212.01539

---

> ### Author Response · Authors · 2024-11-27
> **Follow-up Response**
>
> Upon further reflection, we acknowledge that our initial selection of baseline comparisons may not have been ideally matched, and including work [4] would indeed provide a more relevant comparison than references [1] and [2]. However, we have discovered that slight modifications to the FlashDP algorithm can also support flat clipping. Now we are working on modifying the FlashDP algorithm to support both per-layer and flat-clipping techniques. We are actively conducting experimental validations of these adjustments. We intend to update our manuscript with the revised algorithm and the corresponding experimental results if time permits.
>
> We greatly appreciate the opportunity to engage in this dialogue with you. We are committed to refining FlashDP and making a meaningful contribution to the differential privacy community. If you have any further questions or require additional information about FlashDP and its developments, we are more than happy to continue this conversation and provide the necessary details.

---

### Official Review · Reviewer_aRPR · 2024-11-06

**Soundness:** 3
**Presentation:** 3
**Contribution:** 2
**Rating:** 6
**Confidence:** 4

**Summary:**

This paper studies engineering approaches used to accelerate deep learning and improve the performance of DP-SGD. DP-SGD, the most widely used algorithm for training differentially private machine learning models, is known to cause performance degradation. Specifically, due to the need to clip gradients, initial implementations faced a large memory footprint because of the need to store per-sample gradients. Recent advances have almost solved this problem. A recent method called Ghost clipping eliminates the need to instantiate per-sample gradients, instead instantiating only per-sample norms. However, the throughput of this method is still far from that of non-private training. This paper attempts to resolve this gap by introducing FlashDP, which uses advanced caching techniques and performs gradient norm calculation and gradient calculation in a single task. This eliminates redundant calculations (by 20%, as claimed by the authors) of the activation gradient and improves throughput compared to Ghost clipping. FlashDP achieves a 90% throughput compared to the Non-DP method on a four-A100 system during the pre-training of the Llama-13B model.

**Strengths:**

- Improving the performance of DP-SGD is important for adoption of DP in machine learning community

- Matching the throughput and memory footprint of non-private training is impressive

**Weaknesses:**

- Not much novelty, most techniques seem to be standard engineering techniques to accelerate deep learning

**Questions:**

- Figure 1 isn’t informative, how does eliminating the buffer help the throughput?

- What is subscript T in line 7 of algorithm 1?

- How do you exactly match the performance of DP-SGD? The added noise and sub-sampling creates a variance in the final model but that doesn't show in table 2.

- How do you perform poisson sub-sampling? In my experience, poisson sub-sampling itself can make training slower even without gradient clipping.

- When you fix batch-size, does it mean that you are using sub-sampling rates that lead to that batch-size in expectation?

---

> ### Author Response · Authors · 2024-11-24
> **Response by Authors**
>
> Thank you for your constructive feedback. We appreciate the opportunity to clarify and further explain our approach and findings.
>
> ## For the weaknesses you mentioned:
>
> Reducing memory storage or accelerating DP-SGD has been a hot topic, with contributions like Ghost Clipping and DP-JAX, some of which have appeared at security conferences, such as [1], now integrated into Opacus. Our system-level approach is relatively new to the security community, making this research impactful for a security-focused conference.
>
> [1] Scaling up Differentially Private Deep Learning with Fast Per-Example Gradient Clipping (PoPETS 2021)
>
>
> We would like to highlight that we are the first to integrate several CUDA-related techniques, such as intra- and inter-block shuffle reduction and atomic add, into DP-SGD. These innovations significantly reduce the number of HBM loads and stores and improve the efficiency of DP-SGD, which has historically been bottlenecked by memory movement rather than computation. While these techniques have long been established, adapting them to DP-SGD presents crucial challenges, which we have effectively addressed in our work. **By our approach, for the first time, we make it possible to pre-train a Llama-13B model using DP-SGD with 4 A100 GPU. **
>
> ## For your questions:
>
> 1. Regarding Figure 1: The buffer in Figure 1 is related to GPU High Bandwidth Memory (HBM). By reducing the frequency of HBM accesses, we significantly decrease the latency and overhead associated with memory operations, thus improving throughput.
>
> 2. Clarification of subscript T: In our manuscript, 'T' denotes the sequence length, as described in Section 3.
>
> 3. Matching the performance of DP-SGD: To address the inherent variability introduced by differential privacy, we ensure consistency in our experiments by using the same seeds to generate noise across comparative runs.
>
> 4. In our implementation, we employ subsampling without replacement rather than Poisson subsampling. We opt for fixed-size mini batches because they align well with available GPU memory configurations, which enhances learning throughput. This approach is well-supported by existing literature that investigates the associated privacy budgets, such as the studies found in [[1](https://arxiv.org/pdf/2408.10456)] and [[2](https://arxiv.org/pdf/1808.00087)]. Using subsampling without replacement is a standard practice in scenarios like ours where consistent batch sizes contribute to more efficient computation.
>
> [1] https://arxiv.org/pdf/2408.10456
> [2] https://arxiv.org/pdf/1808.00087

---

### Note · Authors · 2024-12-01

I have read and agree with the venue's withdrawal policy on behalf of myself and my co-authors.